Resource

EMBO
Molecular Medicine

# A broadly applicable protein-polymer adjuvant system for antiviral vaccines

Caiqian Wang[1,2,3,6], Yuanyuan Geng[4,6], Haoran Wang[1,3], Zeheng Ren[1,3], Qingxiu Hou[1,3], An Fang[1,3], Qiong Wu [ID][1,3], Liqin Wu[1,3], Xiujuan Shi[4], Ming Zhou [ID][1,3], Zhen F Fu [ID][1,3], Jonathan F Lovell [ID][5✉], Honglin Jin [ID][4✉] & Ling Zhao [ID][1,2,3✉]

## Abstract

Although protein subunit vaccines generally have acceptable safety profiles with precise antigenic content, limited immunogenicity can lead to unsatisfactory humoral and cellular immunity and the need for vaccine adjuvants and delivery system. Herein, we assess a vaccine adjuvant system comprising Quillaja Saponaria-21(QS-21) and cobalt porphyrin polymeric micelles that enabling the display of His-tagged antigen on its surface. The nanoscale micelles promote antigen uptake and dendritic cell activation to induce robust cytotoxic T lymphocyte response and germinal center formation. Using the recombinant protein antigens from influenza A and rabies virus, the micelle adjuvant system elicited robust antiviral responses and protected mice from lethal challenge. In addition, this system could be combined with other antigens to induce high titers of neutralizing antibodies in models of three highly pathogenic viral pathogens: Ebola virus, Marburg virus, and Nipah virus. Collectively, our results demonstrate this polymeric micelle adjuvant system can be used as a potent nanoplatform for developing antiviral vaccine countermeasures that promote humoral and cellular immunity.

Keywords Nanovaccine; Antigen Delivery; Multivalent Display; Recombinant Protein Antigens; Antiviral Vaccine
Subject Categories Immunology; Methods & Resources; Microbiology, Virology & Host Pathogen Interaction

## Introduction

Vaccination is the most economical and effective measure to prevent infectious diseases. Protein subunit-based vaccines are generally safe, easy to produce, and do not require in vivo translation. However, purified proteins are poorly immunogenic and cannot induce a sufficiently strong and durable protective immune response against lethal pathogens. Nanoparticle-based delivery vehicles offer a promising strategy to overcome the limitations of conventional subunit vaccines (Das and Ali, 2021). Nanovaccine can not only target immune cells, enhance the trafficking of antigens in lymph nodes and facilitate antigen presentation, but also stimulate the immune system using the inherent antigenicity of the vehicle (Alameh et al, 2021; Gheibi Hayat and Darroudi, 2019). In addition, nanotechnology provides advantages in antigen assembly and proportioning, as well as in manipulating antigen density and orientation. Nanocarrier-based protein vaccines have been shown to significantly boost immune responses against pathogens such as SARS-CoV-2, malaria, hepatitis B virus, and human immunodeficiency virus (HIV) (Hanson et al, 2015; Huang et al, 2018; Ma et al, 2020; Wang et al, 2020). Consequently, there is interest to develop new nanoplatforms that are easy to synthesize, have high loading efficiency, stability, and simultaneously promote humoral and cellular responses.

Previous studies have shown that the immune system can recognize unique repetitive structures on the surface of viruses or bacteria (Smith et al, 2013). Repeated antigens display not only can facilitate internalization into antigen-presenting cells (APCs), but also promotes B cell receptor (BCR) co-aggregation, triggering, and activation (Rappuoli and Serruto, 2019; Sun et al, 2023; Zhu et al, 2014). Although self-assembled protein nanoparticles, including ferritin family proteins, pyruvate dehydrogenase (E2), and virus-like particles (VLP) can achieve repeated antigen display, there is a risk of scaffold response immunodominance, displaying antigens in a heterogeneous manner, and masking important epitopes (Feng et al, 2022; Huang et al, 2018). Self-assembled micelles based on amphiphilic polymers such as poly(lactic acid) (PLA) and poly(lactic-co-glycolic acid) (PLGA) are generally reproducible and controllable in terms of biocompatibility, molecular weight composition, targetability, antigen ratio, and degradation rate (Das and Ali, 2021; Perumal et al, 2022). However, the approaches of attaching proteins to polymeric micelles for antigen display are often indirect or ineffective. Although 1,2-Distearoyl-sn-glycero-3-phosphoethanolamine-*N*-[methoxy (polyethylene glycol)-2000] (DSPE-PEG2000)-based phospholipid micelles have been

[1]National Key Laboratory of Agricultural Microbiology, Huazhong Agricultural University, Wuhan 430070, China. [2]Hubei Hongshan Laboratory, Wuhan 430070, China. [3]Key Laboratory of Preventive Veterinary Medicine of Hubei Province, College of Veterinary Medicine, Huazhong Agricultural University, Wuhan 430070, China. [4]College of Biomedicine and Health and College of Life Science and Technology, Huazhong Agricultural University, Wuhan 430070, China. [5]Department of Biomedical Engineering, University at Buffalo, State University of New York, Buffalo, NY 14260, USA. [6]These authors contributed equally: Caiqian Wang, Yuanyuan Geng. ✉E-mail: jflovell@buffalo.edu; jin@hust.edu.cn; lingzhao@mail.hzau.edu.cn

successfully used for the delivery of therapeutic peptides, anchoring large molecular weight proteins that fully display antigenic epitopes is still difficult (Esparza et al, 2019). Our previous research has found that a cobalt porphyrin-phospholipid bilayer can effectively capture his-tagged proteins by utilizing the chelation of histidine and metal ion $Co^{2+}$, enabling antigens to be densely displayed on its surface (Shao et al, 2015). Nevertheless, it has not been reported whether the combination of single-layer polymeric micelles and cobalt porphyrins can achieve the efficient attachment of soluble proteins. Compared to liposomes, micelles are typically more straightforward to formulate and have large-scale production.

Adjuvants are key components in enhancing the breadth and persistence of immune responses to vaccines (Pulendran et al, 2021). Saponins, a promising adjuvant class that induces protective cellular immunity, have been approved for use in multiple human vaccines (den Brok et al, 2016). Quillaja Saponaria-21 (QS-21) is a saponin adjuvant fraction with low toxicity and relatively high yield (Pifferi et al, 2021). The liposomal formulation (AS01) containing monophosphoryl lipid A (MPLA) and QS-21 is a component of GSK's advanced or licensed malaria (Mosquirix), shingles (Shingrix), and RSV (Arexvy) vaccines (Romerio et al, 2023). Previous research has shown that QS-21 efficiently generates antigen-specific $CD8^+$ T cell and antibody responses compared to classical vaccine adjuvants such as aluminum hydroxide (alum) (Welsby et al, 2016). Moreover, QS-21 can elicit humoral responses of all IgG isotypes (IgG1, IgG2, and IgG2b) with a mixed Th1/Th2 balance (Pifferi et al, 2021). The fragment crystallizable (Fc) domain is one part of the antibody molecule that interacts with Fc-receptors (FcRs) (Czajkowsky et al, 2012). Due to the interaction of Fc with neonatal Fc-receptors (FcRn) and the slower renal clearance rate of larger-sized molecules, the presence of the Fc domain significantly increases the plasma half-life of its fusion proteins (Kontermann, 2011; Roopenian and Akilesh, 2007). In addition, the Fc domain can also promote the interaction between the fused protein and the Fc-receptors (FcRs) in immune cells (Delidakis et al, 2022). Although previous studies have shown that crystallizable Fc-fusion dimers can enhance the immunogenicity of recombinant proteins (Liu et al, 2020; Tai et al, 2022), it remains unclear whether the incorporation of Fc-fusion proteins into polymeric micelles can further enhance the immunogenicity of the protein.

Here, we combined the advantages of the high stability of polymeric micelles and the ability of cobalt porphyrin to chelate His-tagged proteins, constructing a novel nanocarrier platform based on PLA-Porphyrin-$Co^{2+}$ and DSPE-PEG$_{2000}$, termed PPCD. This strategy enables the self-assembly and dense display of macromolecular antigens (Fc-fused proteins) on the PPCD surface, as well as the QS-21 adjuvant payload (termed PPCDQ), achieving co-delivery of antigen and adjuvant (Fig. EV1). Furthermore, we evaluated the effectiveness of this nanoplatform in mice by using the hemagglutinin (HA) protein of influenza A virus (IAV) and the rabies virus (RABV) glycoprotein (RABV-G) as models. We found that both the nanovaccines (HA@PPCDQ and RABV-G@PPCDQ) elicited higher neutralizing antibodies than other protein subunit vaccine formulations and protected mice against lethal IAV and RABV challenges. Meanwhile, the universality of the PPCDQ nanoplatform was further verified with the antigens of the Ebola virus (EBOV), Marburg virus (MARV), and Nipah virus (NiV). Further experiments showed that HA@PPCDQ could promote antigen residence time in lymph nodes and antigen uptake by APCs, as well as trigger robust germinal center responses and T-cell responses. Overall, the PPCDQ protein-polymer adjuvant system notably increases the immunogenicity of subunit proteins through lymph node targeting, sustained antigen release, and DC activation.

# Results

## Construction and characterization of HA@PPCDQ

To construct the PPCD nanocarrier, PLA-Porphyrin-$Co^{2+}$ (synthetic route shown in Appendix Fig. S1) was mixed with 1,2-Distearoyl-$sn$-glycero-3-phosphoethanolamine-$N$-[methoxy (polyethylene glycol)-2000] (DSPE-PEG2000) in equal proportions and then added dropwise to the phosphate buffer saline (PBS) solution. To further evaluate the applicability of the PPCD nanocarrier platform and explore whether the fusion of the Fc domain would enhance the immune effect, we used the Expi 293 F cell system to express three types of HA proteins of IAV (C-His-HA, C-His-Fc-HA, and N-His-Fc-HA). The plasmid construction strategy was shown in Fig. EV2A, CMV was used as a promoter, a IgG1 signal peptide (SP) was fused to the N-terminal, an 8-histidine ($8 \times$ His) tag was fused to the N-terminal or C-terminal, and Fc domain was fused to the C-terminal of HA. All proteins were purified by affinity chromatography and verified by SDS–polyacrylamide gel electrophoresis and Coomassie brilliant blue staining (Fig. EV2B). Finally, PPCD nanocarrier, QS-21 adjuvant, and His-tagged HA protein were mixed and incubated in PBS buffer to self-assemble into PPCDQ nanovaccine (Fig. 1A). The PPCD loaded with QS-21 is termed PPCDQ, abbreviated as NP. The PPCDQ micelles combined with different HA proteins are hereinafter referred to as His-HA-NPs, C-His-Fc-HA-NPs, and N-His-Fc-HA-NPs (Fig. 1B), and HA@PPCDQ is the general term for these three NPs.

Native polyacrylamide gel electrophoresis (PAGE) was used to detect the anchoring efficiency of PPCD micelles to His-tagged HA protein. We found that 1 mg of PPCD micelles can bind at least 0.4 mg of HA protein, that is, a 2.5:1 mass ratio of PPCD to protein was sufficient for binding (Fig. EV2C,D). This data suggests that PPCD micelles have superior loading capacity for His-tagged proteins. Meanwhile, PPCD micelles can also bind well to larger molecular weight proteins fused with the Fc domain (C-His-Fc-HA and N-His-Fc-HA, Fig. EV2E,F). Size-exclusion chromatography (SEC) further confirmed this result (Appendix Fig. S2). In addition, cobalt-deficient PPDQ micelles and non-His-tagged Fc-HA protein was utilized to confirm cobalt/His-tag mediated association. Our results showed cobalt-deficient PPDQ micelles cannot anchor His-HA protein, and PPCDQ cannot capture Fc-HA protein lacking His-tag, suggesting that the conjugation between PPCDQ micelles and antigens is mediated by cobalt and His-tag (Appendix Fig. S3). Next, we measured the physicochemical properties of HA@PPCDQ, the data of transmission electron microscopy (TEM) clearly showed that HA@PPCDQ were spherical nanoparticles and the surface of the NPs was smooth, while there are many spots on the surface of His-HA-NPs, C-His-Fc-HA-NPs, and N-His-Fc-HA-NPs, which are presumed to be HA antigens loaded on the micelle surface (Fig. 1C). Dynamic light scattering (DLS) data showed a hydrodynamic diameter of $111 \pm 9.7$ nm for NPs, $113 \pm 8.4$ nm for His-HA-NPs, $119 \pm 5.8$ for C-His-Fc-HA-NPs, and $116 \pm 9.7$ nm for N-His-Fc-HA-NPs (Figs. 1D and EV2G).

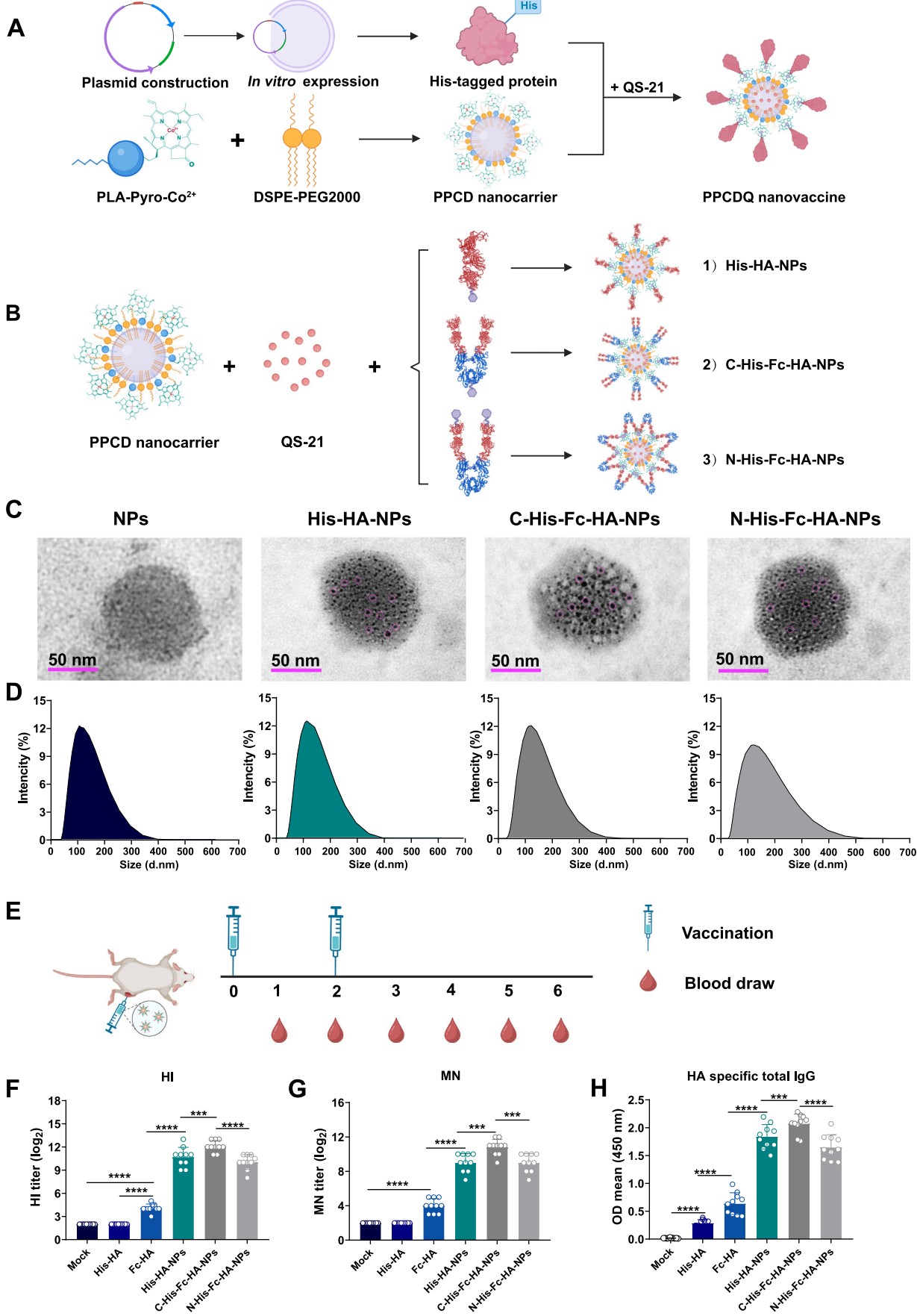

**Figure 1.  Construction, characterization, and immunization of HA@PPCDQ.**

(A) Schematic illustration of the preparation process of PPCDQ nanovaccine (The positions of all materials in the diagram are relative positions drawn to display the antigens). (B) The procedure of HA@PPCDQ production. PPCD nanocarrier, QS-21, and three kinds of His-tagged HA protein were mixed and incubated in PBS buffer to self-assemble into HA@PPCDQ nanovaccines (His-HA-NPs, C-His-Fc-HA-NPs, and N-His-Fc-HA-NPs). (C) TEM images of PPCDQ (NPs), His-HA-NPs, C-His-Fc-HA-NPs, and N-His-Fc-HA-NPs. The rose-red circles in the picture are part of the protein bound to the surface of the micelle. Scale bars: 50 nm. (D) Hydrodynamic diameter distribution of each nanoparticle ($n = 3$ biological replicates per group). (E) Schematic of C57/BL6 vaccination. Mice from each group were prime/boost-vaccinated (I.M.) with different vaccines at week 0 and week 2. Serum was collected every week. All mice were euthanized at week 6. (F–H) Hemagglutination inhibition (HI) titers (F), microneutralization (MN) titers (G), and HA-specific total IgG (H) at 3 weeks after immunization ($n = 10$ animals per group). Data information: Data in (F–H) are presented as mean ± SD. Adjusted $P$ values were calculated by one-way ANOVA with Tukey's multiple comparisons test. ***$P < 0.001$, ****$P < 0.0001$. The experiments shown are representative of at least two independent experiments. Source data are available online for this figure.

The zeta potential of HA@PPCDQ was measured by Malvern laser Doppler velocimetry and the result indicated antigen binding altered the particle surface to less electronegative (Fig. EV2H). Additionally, in vitro hemolysis assay showed that the three micelles did not cause hemolysis of red blood cells at a concentration of 250 μg/mL (the maximum concentration for in vivo experiments), and did not cause cytotoxicity on DC2.4 cells (Appendix Fig. S4), indicating that PPCDQ micelles have a favorable safety profile and beneficial for the in vivo utility. Taken together, these results indicate that we successfully constructed three nanovaccines (His-HA-NPs, C-His-Fc-HA-NPs, and N-His-Fc-HA-NPs) against IAV using the PPCDQ nanocarrier system.

## HA@PPCDQ elicits robust humoral immune responses

To evaluate the immunogenicity of HA@PPCDQ, C57/BL6 mice were intramuscularly immunized with three different nanovaccines containing 20 μg His-HA, C-His-Fc-HA, and N-His-Fc-HA or the same mass of the soluble protein. The immunization scheme is shown in Fig. 1E; mice were primed at week 0 and booster at week 2. Serum was collected every week and assessed with hemagglutination inhibition (HI), microneutralization (MN) and antigen-specific ELISA assay. As shown in Fig. 1F–H, all three nanovaccines induced significantly higher antibody response than soluble proteins. In particular, the HI titer induced by C-His-Fc-HA-NPs was 256-fold higher than Fc-HA at 3 weeks after immunization (Fig. 1F). The HI and MN titer reached a peak at 4 weeks post-immunization (w.p.i.) while HA-specific IgG reached a peak at 5 w.p.i. (Fig. EV3A–C). Mice in HA monomers-vaccinated groups only induced HA-specific IgG, the HI and MN titers were undetectable (Fig. 1F–H). About 30% of mice in HA-Fc dimers-vaccinated groups induced protective antibodies above the limit of detection (HI titer >4), indicating that the HA dimer was more immunogenic than the monomer. The antibody response of C-His-HA-Fc-NPs was significantly higher than His-HA-NPs at week 3. However, N-His-HA-Fc-NPs had comparable antibody levels compared to His-HA-NPs, and antibody growth was slow during the first two weeks after primary immunization (Figs. 1F–H and EV3A–C). Therefore, the follow-up immunological experiments mainly focus on His-HA-NPs and C-His-Fc-HA-NPs. In general, our results indicate that the PPCDQ platform combined with all three HA proteins can elicit notable antibody responses and greatly enhance the immunogenicity of the protein vaccine.

## Synergy of PPCD and QS-21 in antibody production

To further identify whether the enhanced antibody response has an antigen dose-dependent effect, we compared His-HA-NPs and

C-His-Fc-HA-NPs at doses of 5, 10, and 20 μg. Unexpectedly, there was no significant difference in antibody response among the three doses (Fig. EV3D,E). On the contrary, the immune effect was positively related to the dose of QS-21 (Fig. EV3I). Meanwhile, we evaluated the influence of intramuscular injection (I.M., on both sides of the hind limbs) and subcutaneous injection (S.Q., at the base of the tail) on the immune effect of His-HA-NPs. As shown in Fig. EV3F, mice vaccinated by I.M. produced a higher titer of antibody at 3 weeks post-immunization compared with S.Q. administration and no significant difference was detected at other time points. In addition, the effects of different micelles and the presence or absence of QS-21 adjuvant on immune efficacy was also examined when the antigen type and dose were the same (Fig. EV3G). Surprisingly, we found immunization with a mixture of His-HA and PPD micelles induced detectable anti-HA antibodies (HI titer >4), speculating that the inherent immunogenicity of PPD micelles induced some degree of immune response (Alameh et al, 2021; Gheibi Hayat and Darroudi, 2019). Likewise, enhanced immunogenicity was also detected when His-HA was administered with PPCD lacking QS-21, which is likely due to the repeated antigen display and carrier effect of PPCD (Smith et al, 2013; Wang et al, 2020). However, the enhancement became more notable when the QS-21 was present, suggesting a synergistic effect between antigen display by PPCD nanocarrier and adjuvant-induced innate response. Moreover, we found that His-HA@PPDQ (cobalt deficient) and Fc-HA@PPCDQ (lack of His tag) induced lower HI titers compared to His-HA@PPCDQ and His-Fc-HA@PPCDQ (Appendix Fig. S3), further illustrating that antigen display can enhance the immunogenicity of proteins. Taken together, these results demonstrate that the robust humoral response elicited by HA@PPCDQ is caused by the synergistic effect of PPCD nanocarrier and QS-21 adjuvant.

Additional commercial adjuvants, including Alum, AddaVax (similar to MF59), and AddaS03 (a AS03-like vaccine adjuvant) were compared with PPCDQ. Among them, MF59 and AS03 have been used in approved influenza vaccines (Roth et al, 2022). As shown in Fig. EV3H, all three commercial adjuvants mixed with HA@PPCD induced higher HI titers compared to the mixture with HA monomers. However, the antibody titers were still significantly lower than those of His-HA-NPs, approximately three times lower. Due to the commercial adjuvant itself is in the form of nanoparticles (according to the manufacturer's instructions), it may not be able to integrate with PPCD to function in the same peripheral lymph nodes or cells. Besides, we found there is no significant difference in antibody levels between His-HA-NPs and C-His-Fc-HA-NPs at high doses, which may be caused by saturation of antigen density. We further evaluated the antibody

responses of His-HA-NPs and C-His-Fc-HA-NPs at 1 μg dose. The results showed that both His-HA-NPs and C-His-Fc-HA-NPs produced high titers of protective antibodies to inhibit IAV even at low doses, and the antibody titers produced by C-His-Fc-HA-NPs approximately 2.6-fold higher than that of His-HA-NPs (Fig. EV3J). Thus, Fc-fused protein may be a more optimal choice from the standpoint of saving antigens.

## HA@PPCDQ causes efficient antigen uptake and sustained antigen release

Our above results indicate that HA@PPCDQ induces a stronger immune response compared to other adjuvant systems. To elucidate whether potent antibody responses benefit from efficient antigen uptake and processing, we prepared Cy5-tagged, Cy7-tagged, iFlour™ 594-tagged, and iFlour™ 488-tagged HA to track the antigen. First, we compared whether there are differences in lymph node accumulation and antigen-presenting cell uptake between the Fc-HA-NPs and His-HA-NPs. Two Cy5-labeled antigens (Cy5-His-HA and Cy5-His-Fc-HA) were used to tracing and the results of vivo imaging and flow cytometry showed there is no significant difference between the two micelles (Fig. EV4A–D; Appendix Fig. S5). Therefore, we chose His-HA and His-HA-NPs for subsequent experiments, focusing on comparing the differences between monovalent and multivalent antigens in lymph node accumulation, APC uptake, and presentation. C57/BL6 mice were immunized with Cy5-tagged His-HA-NPs or an equivalent amount of Cy5-tagged HA, and inguinal lymph nodes (iLNs) were harvested at 4 h post vaccination for FCM analysis and living image. We found that the percentages of HA$^+$ DCs (B220$^-$CD11c$^+$MHC-II$^+$Cy5$^+$) and macrophages (B220$^-$CD11b$^+$F4/80$^-$CD169$^+$ Cy5$^+$) in His-HA-NPs-vaccinated mice were significantly higher than those in HA-vaccinated mice (Fig. 2A,C,D). The data of living image showed that more Cy7-tagged His-HA-NPs accumulated in lymph nodes compared to His-HA protein (Fig. EV4E,F), implying that HA@PPCDQ were easier targeted to LNs and captured by DCs and macrophages preferentially. These results were further verified by antigen uptake experiments *in vitro*. As expected, the mean fluorescence intensity (MFI) of HA in cells increased around threefold when DC2.4 and RAW264.7 cells were treated with iFlour™ 594-tagged His-HA-NPs (Fig. 2B,E). Furthermore, we found more His-HA-NPs co-localized with lysosomes, indicating that more internalized antigens are degraded into peptides for subsequent loading onto MHCII molecules.

Previous reports have shown that nanovaccines can be efficiently transported in the lymphatic system and accumulate in lymph nodes to enhance immune processing (Chattopadhyay et al, 2017; Ramos-Gomes et al, 2020; Zhang et al, 2019). To track the flow of HA@PPCDQ in vivo after immunization, we constructed the iFlour™ 594-tagged His-HA-NPs and immunized the CD11c-EYFP transgenic mice by hindlimb intramuscular injection. As shown in Fig. 2F, His-HA-NPs were significantly co-localized with the CD11c$^+$ DCs in the medullary region of inguinal lymph nodes at 4 h and 7 days post vaccination. In contrast, the Flour™ 594-tagged HA was barely observable in the protein group at 7 days after immunization, indicating that PPCDQ can prolong the antigen residence time in lymph nodes. We also observed the distribution of antigens in other lymph nodes in the body and found His-HA-NPs can not only accumulate to the lymph nodes

near the injection site (inguinal, popliteal and mesenteric LNs), but also rapidly flow to the distant lymph nodes (axillary and cervical LNs, Fig. EV4G). To further investigate the spatial location of His-HA-NPs in lymph nodes after immunization, we immunized C57/BL6 mice with iFlour™ 488-labeled His-HA-NPs. Lymph nodes were harvested at 4 h after vaccination for light sheet microscopy scanning. The images of whole LNs revealed that His-HA-NPs was primarily distributed in the subcapsular sinus and collagen conduits of the inguinal lymph node (Fig. 2G; Movie EV1) and the subcapsular sinus of the axillary lymph node (Fig. 2H; Movie EV2).

The large size of multivalent nanoparticles has been shown to affect their transport *in vivo*, effectively slowing down their accumulation kinetics in LNs, thereby enhancing antigen retention and improving immunogenicity (Martin et al, 2020; Ols et al, 2023). In addition, slow antigen delivery can enhance neutralizing antibody and germinal center responses (Cirelli et al, 2019; Tam et al, 2016). To investigate whether the PPCDQ system can delay antigen release, we used the IVIS® spectrum in vivo imaging system (PerkinElmer) to monitor antigen retention at the injection site in mice for 12 consecutive days. Intravital imaging revealed that pure protein vaccines have almost no detectable fluorescent signal at 9 dpi (Fig. 2I,J). However, the average radiant efficiency in the His-HA-NPs group was over 10$^7$ ROI at 12 dpi. In short, our results indicate that PPCDQ induces effective antigen uptake and prolongs antigen retention time at the injection site and lymph node.

## HA@PPCDQ promotes DC activation and depletes CD169$^+$ macrophages

DCs are the bridge connecting innate immunity and adaptive immunity. Activated DCs are essential for naive T cells to become arming effector T cells and to trigger their effector functions to attack pathogen-infected cells (Hilligan and Ronchese, 2020; Patente et al, 2019; Walsh and Mills, 2013). To explore the effect of HA@PPCDQ on DCs, we examined the markers of DC activation in vitro and in vivo by flow cytometry (FCM) analysis. The results in Fig. 3A,B showed that MHC-II, CD80, and CD86 molecules were significantly upregulated after HA@PPCDQ treated BMDCs for 12 h. Consistent with the results of in vitro studies, more DCs (B220$^-$CD11c$^+$) expressed MHC-II, CD80, and CD86 were detected in HA@PPCDQ-vaccinated group (Figs. 3C and EV5A), indicating that HA@PPCDQ can promote DC maturation in vitro and in vivo. Subcapsular sinus macrophages (SSMs) are "sentinel" cells that capture pathogens entering lymph nodes, they can directly drag surface-bound antigens into the follicle, which allows B cells to encounter them or localizing them on the surface of follicular dendritic cells (FDCs) (Louie and Liao, 2019; Moran et al, 2019). Previous studies have shown that QS-21 rapidly accumulated in CD169$^+$ SSM after intramuscular injection and led to rapid loss of SSM (Detienne et al, 2016). As previously demonstrated, we found that the number of SSMs (B220$^-$CD11b$^+$F4/80$^-$CD169$^+$) in the HA@PPCDQ-vaccinated group decreased about three times compared with those in a protein-vaccinated group at 2 days after booster immunization. In addition, the percentage of medullary sinus macrophages (MSMs) also declined (Fig. 3D; EV5B). Immunofluorescence staining of draining lymph nodes also confirmed this result (Fig. EV5F). To further verify whether the QS-21 loaded in the HA@PPCDQ

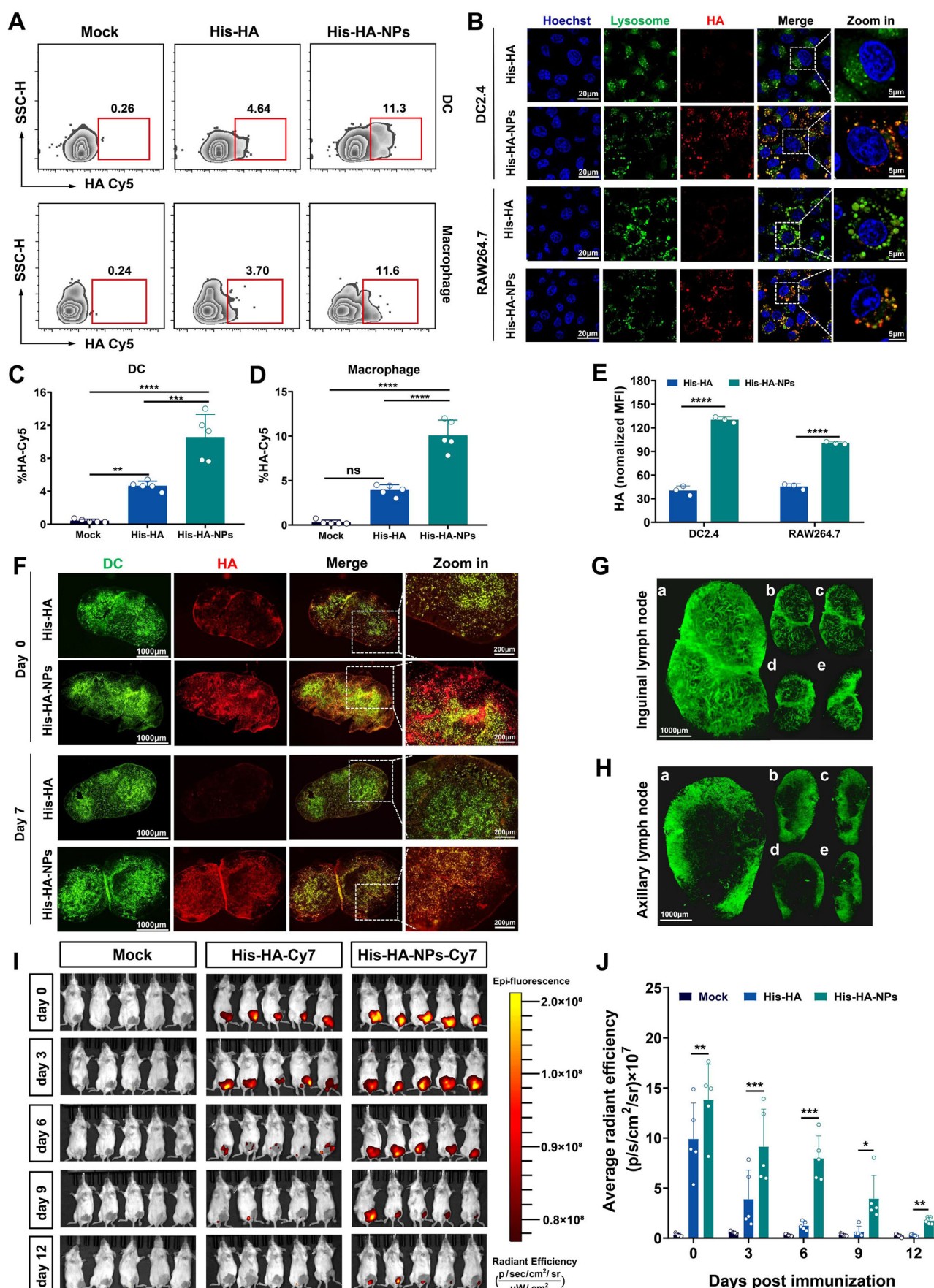

◀ **Figure 2. HA@PPCDQ facilitates antigen uptake and delays antigen release.**

(A) Frequencies of Cy5 positive DCs (B220⁻CD11c⁺MHC-II⁺, top panel) and macrophages (B220⁻CD11b⁺F4/80⁺, bottom panel) in iLNs at 4 h post injection. (B) Antigen uptake experiment of DCs and macrophages in vitro. (B) represented the distribution of HA (red) and lysosome (green) in RAW264.7 and DC2.4 cells after incubation of equivalent iFlour™ 594-tagged His-HA or His-HA-NPs for 12 h. The blue staining indicated Hoechst-stained nuclei. Scalar bar: 20 μm (left), 5 μm (right). (C) Statistical graphs of Cy5 positive DCs ($n = 5$ animals per group). (D) Statistical graphs of Cy5 positive macrophages ($n = 5$ animals per group). (E) Statistical graphs of the HA mean fluorescence intensity (MFI) analyzed by ImageJ software ($n = 3$ biological replicates per group). (F) The distribution of iFlour™ 594-tagged His-HA or His-HA-NPs in inguinal lymph nodes at day 0 and day 7 post injection. (G, H) Light sheet microscopy visualizing iFlour™ 488-tagged His-HA-NPs in inguinal lymph node (G) and axillary lymph node (H) at 4 h post injection. (a-e) represented the different angles of lymph nodes. 360° views are available in Movie EV1 and Movie EV2. (I) The residency of Cy7-tagged His-HA and His-HA-NPs at the injection site was observed using an IVIS Spectrum system. p, photon. ($n = 5$ animals per group). (J) Statistical graphs of the average radiant efficiency analyzed by Living Image Vision 4.4 ($n = 5$ animals per group). Data information: Data in (C–E, J) are mean ± SD. Statistical significance was calculated by one-way ANOVA with Tukey's multiple comparisons test in (C, D) and Two-way ANOVA with Tukey's multiple comparisons test in (E, J). *$P < 0.05$, **$P < 0.01$, ***$P < 0.001$, ****$P < 0.0001$. Source data are available online for this figure.

system is the cause of SSM depletion, we tested the SSM levels after immunization with His-HA + QS-21 and His-HA + PPCD. As shown in Fig. EV5C–E, immunization with His-HA or His-HA + PPCD did not lead to SSM depletion, while immunization with His-HA + QS-21 and His-HA-NPs led to the same degree of SSM depletion.

Due to SSMs and the marginal-zone metallophilic macrophages (MMMs) lining the marginal sinus of the spleen have similar phenotypes and ontogeny (Moran et al, 2019). We further examined the effect of HA@PPCDQ immunization on spleen macrophages. We found no difference in the percentage of tissue-resident macrophages (F4/80ʰⁱCD11bⁱⁿᵗ) and monocyte-derived macrophages (F4/80⁻CD11bʰⁱ) among the different immune groups (Fig. 3E,G). In contrast, the number of neutrophils in the HA-Fc-NP immunized group was significantly increased, about ten times higher than that in the control group, implying that HA@PPCDQ promotes the recruitment of neutrophils to the spleen after immunization. Consistent with the phenotype of SSMs in lymph nodes, the frequency of CD169⁺ cells (MMMs) in the tissue-resident macrophage population were notably reduced (Fig. 3F,H). Whereas, the proportion of CD169⁺ cells increased in the neutrophil population and remained unchanged in the monocyte-derived macrophages population. Collectively, our results demonstrated that HA@PPCDQ facilitates DC activation in vitro and in vivo, and depletes the CD169⁺ tissue-resident macrophages in lymph nodes and spleen.

## HA@PPCDQ drives both Th1 and Th2 cell response in mice

Naive T cells are initially activated with the recognition of antigenic peptides presented by APCs through the T cell receptor (TCR). And further activated by co-stimulatory receptor CD28 cross-linking with B7 family molecules CD80/CD86 (ligands) on the surface of APCs (Summers deLuca and Gommerman, 2012). Our results above suggest that HA@PPCDQ can rapidly accumulate in lymph nodes and be taken up by DCs and macrophages. Therefore, the activation of T cells after HA@PPCDQ immunization was further examined. We found that the early activation marker of T cells, CD69, upregulated about tenfold in both CD4⁺ and CD8⁺ T cells in HA@PPCDQ immunized mice (Fig. 4B,E; Appendix Fig. S7). And the HA@PPCDQ group induced more CD4⁺ and CD8⁺ effector memory T (TEM, CD62L⁻CD44⁺) cells (Fig. 4A,B,E). In addition, the frequency of CD4⁺ and CD8⁺ central memory T cells (TCM, CD62L⁺CD44⁺) in HA@PPCDQ group were significantly higher than those in the protein group (Fig. 4B–E).

In short, these results implied that HA@PPCDQ can activate T cells remarkably.

QS-21 was the core component of HA@PPCDQ to significantly enhance the immunogenicity of antigens. Previous studies have shown that QS-21 stimulates both antibody-based and cell-mediated immune responses, primarily eliciting a Th1-biased immune response (Kensil, 1996; Pifferi et al, 2021). In order to determine whether HA@PPCDQ induced immune bias, we analyzed the IFN-γ⁺ and IL-4⁺ producing cells in splenocytes by both the enzyme-linked immune absorbent spot (ELISpot) and intracellular cytokine staining (ICCS) assays. As shown in Fig. 4F,G, IFN-γ producing cells increased about sevenfold in the HA@PPCDQ group, while the levels of IL-4-producing cells were also notably affected, increasing about 20-fold. Consistent with the ELISpot results, the frequency of both antigen-specific IFN-γ⁺ CD4⁺ T cells (Th1-biased cells) and IL-4⁺ CD4⁺ T cells (Th2-biased cells) in the HA@PPCDQ group were significantly higher than those in the protein group (Fig. 4H; Appendix Fig. S8), suggesting that HA@PPCDQ drives both Th1 and Th2 cell responses.

The enhanced T-cell responses were further confirmed by antigen-specific ELISA assay. We found that IgG2a subtype is predominant, IgG1 and IgG2b were equivalent (Appendix Fig. S11). Evaluating the ratio of the three subtypes, HA@PPCDQ elicited higher Th1-biased immune responses. It has been reported that TNF-α-producing CD4⁺ T cells are associated with durable antibody responses (van der Ploeg et al, 2022). We found that NPs-vaccinated group generated five times more TNF-α⁺ CD4⁺ T cells than the His-HA or Fc-HA group (Fig. 4H; Appendix Fig. S8). Moreover, HA@PPCDQ induced more CD4⁺TNF-α⁺IL-2⁺IFN-γ⁻ T cells, and there is a strong correlation between this population and HA-specific IgG titer in Fc-HA, His-HA-NPs, and Fc-HA-NPs group(Appendix Fig. S9). Cytotoxic T lymphocytes (CTL cells) are key components of the adaptive immune system, which playing a vital role in defense against pathogens(Fazilleau et al, 2009). Our data showed that HA@PPCDQ induced more antigen-specific, polyfunctional CD8⁺ T cells expressing IFN-γ, IL-2, and TNF-α than the protein vaccines (Fig. 4I; Appendix Fig. S8). Overall, these results indicate that HA@PPCDQ drives robust Th1, Th2, and CTL cell responses.

## HA@PPCDQ induces germinal center responses

The germinal center (GC) is a site for B cell clonal expansion and affinity maturation that leads to the production of high-affinity antibodies (Hägglöf et al, 2023). To determine the effect of

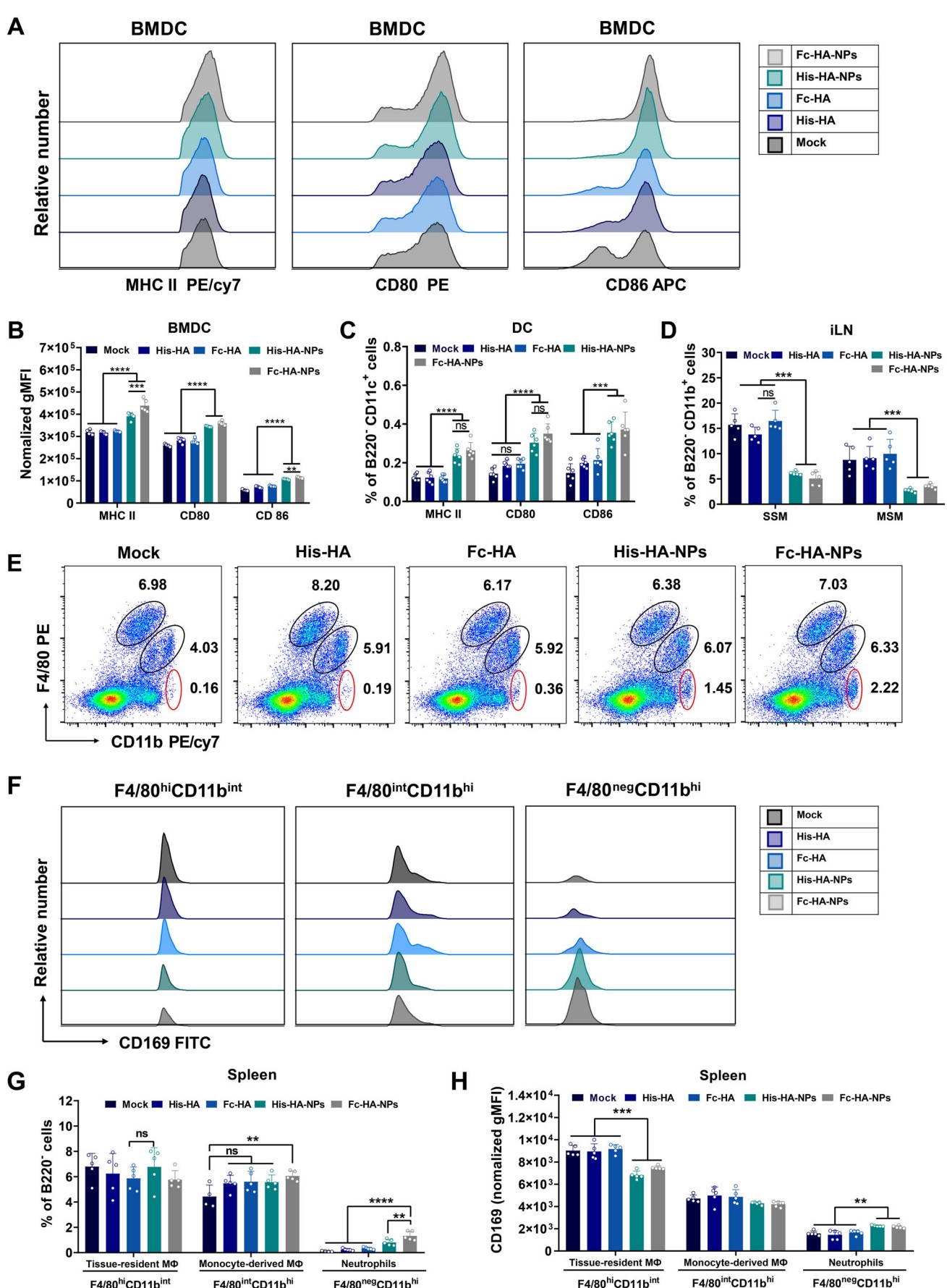

◀ **Figure 3. HA@PPCDQ promotes DC activation and depletes CD169+ macrophages.**

(A) FCM analysis of DC activation markers MHC-II, CD80, and CD86. (B) Represented the statistical graphs of the normalized MFI of MHC-II, CD80, and CD86 ($n = 5$ biological replicates per group). (C) FCM analysis to determine the percentages of CD11c+MHC-II+DCs, CD11c+CD80+ DCs, and CD11c+CD86+DCs in iLN at 2 days post boost immunization ($n = 6$ animals per group). (D) Statistical results of subcapsular sinus macrophage (SSM, B220-CD11b+F4/80-CD169+) and medullary sinus macrophage (MSM, B220-CD11b+F4/80+CD169+) in iLN at 2 days post boost immunization ($n = 5$ animals per group). (E) Spleens at 2 days post boost immunization were processed into single cells for FCM analysis, B220- non-B cells were identified into three distinct populations: tissue-resident macrophage (MΦ, B220-F4/80hiCD11bint), monocyte-derived MΦ (B220-F4/80intCD11bhi) and neutrophil (B220-F4/80-CD11bhi). (F) CD169 expression of tissue-resident MΦ, monocyte-derived MΦ, and neutrophils. (G) Statistical results of tissue-resident MΦ, monocyte-derived MΦ, and neutrophils in the spleen ($n = 5$ animals per group). (H) Statistical results of the normalized MFI of CD169 ($n = 5$ animals per group). Data information: Data in (B–D, G, H) are mean ± SD. Adjusted $P$ values were calculated by one-way ANOVA with Tukey's multiple comparisons test. **$P < 0.01$, ***$P < 0.001$, ****$P < 0.0001$. Source data are available online for this figure.

HA@PPCDQ on GC response, we evaluated the percentage of GC B cells by FCM. We found about ten times more GC B (CD95+GL7+) and HA-specific GC B (CD95+GL7+HA+) cells were detected in the HA@PPCDQ group compared with the His-HA or Fc-HA group (Fig. 5A–C; Appendix Fig. S12A). In addition, we quantified the number of GC in the inguinal lymph nodes (iLNs) at 1 week after the booster immunization using immunofluorescence staining. As shown in Fig. 5J,L, the number of GCs in the HA@PPCDQ group was four to eight times higher than that of the His-HA or Fc-HA group. Follicular helper T (Tfh) cells are critical for the affinity selection of GC B cells. They secrete cytokines IL-21 and IL-4, which work together with the co-stimulatory molecule CD40L to provide the signals required for B cell proliferation and differentiation (Crotty, 2015; Streeck et al, 2013). Cytokine-independent activation-induced marker (AIM) assay has been developed for antigen-specific GC Tfh cells in lymphoid tissue (Dan et al, 2016; Nguyen et al, 2023; Reiss et al, 2017). Our results showed that HA@PPCDQ induced a higher frequency of GC Tfh (CXCR5+PD1+) and HA-specific GC Tfh (CD25+OX40+) cells (Fig. 5D–F; Appendix Fig. S12B). After the B cells that have completed affinity saturation and subtype conversion leave GC, some differentiate into long-lived memory B cells (MBCs), some differentiate into plasma cells (PCs) that partially migrate to the bone marrow to develop long-lived plasma cells (LLPCs) (Laidlaw and Ellebedy, 2022; Young and Brink, 2021). To assess whether HA@PPCDQ immunization affects B cell differentiation, we examined the B cell subsets in iLNs at 2 weeks after boost vaccination. We found HA@PPCDQ generated more HA-specific switched MBCs (IgD-CD38+HA+) compared to protein groups(Fig. 5G–I; Appendix Fig. S12C), and about seven times more LLPCs (B220loCD138+) were detected (Fig. 5O; Appendix Fig. S12D). The antibody-secreting cells (ASCs) in iLN and bone marrow were further quantized by ELISpot assay. As expected, the number of HA-specific ASCs in the His-HA or Fc-HA group did not increase to the same extent after immunization (about eightfold lower) when compared with those in NPs group (Fig. 5K,M,N). Notably, His-FC-HA-NPs induced a more robust B cell response than His-HA-NPs, mainly in the induction of more antigen-specific GC B cells, Tfh cells, and switched MBCs. Based on the findings above, we conclude that HA@PPCDQ elicits potent germinal center response by generating more HA-specific GC B and Tfh cells, and provide persistent protective humoral responses.

## HA@PPCDQ protects mice against lethal IAV challenge

In order to assess the immune effects of HA@PPCDQ in vivo, mice were challenged with $1.8 \times 10^4$ TCID$_{50}$ of influenza A/Puerto Rico/

8/1934 (H1N1) virus at 4 weeks after booster vaccination (Fig. 6A). A 20 μg dose of HA@PPCDQ could protect all mice from lethal IAV challenge (Fig. 6C). In contrast, all mice in the control and pure protein groups died 5–8 days post challenge. In terms of body weight, all HA@PPCDQ-vaccinated mice lost weight slightly in the first 3 days, and began to return to normal levels in the next few days, while protein-vaccinated mice lost weight drastically (Fig. 6B). HA@PPCDQ at doses of 5 μg and 10 μg also protected all mice against lethal IAV virus challenge (Appendix Fig. 16). Histopathological analysis of lungs revealed that IAV challenge induced severe lung lesions in groups immunized protein, which were characterized by thickened alveolar septa, severe bronchiolar necrosis, pulmonary edema, and infiltration of inflammatory cells (Fig. 6D, top). However, none of the mice in the HA@PPCDQ-vaccinated group showed pathological changes. Indirect immunofluorescence assays (IFA) against IAV nucleoprotein (NP, Influenza A) showed dense distribution of NP-expressing cells in the lungs of mice vaccinated with pure protein or placebo, which was not observed in HA@PPCDQ immunized mice (Fig. 6D, bottom). Altogether, these findings suggest that vaccination with HA@PPCDQ protects the mice against lethal IAV infection, mainly through efficient virus clearance, and reduced lung inflammation.

To evaluate whether HA@PPCDQ causes systemic or local toxicity, we isolated hearts, livers, spleens, kidneys, lymph nodes, brains, and small intestines of mice for hematoxylin and eosin staining 1 week after the booster immunization. As shown in Appendix Fig. 17, we found that no lesion was seen in any tissue of the mice immunized with His-HA, Fc-HA, His-HA-NPs, and Fc-HA-NPs compared with the control group, indicating that vaccination with HA@PPCDQ nanoparticle is potentially safe in vivo. In addition, we assessed the stabilization time of PPCDQ micelle pre-bound with His-Fc-HA protein at 4 °C. We found that Fc-HA-NPs can be stored at 4 °C for at least 3 weeks and maintain immunogenicity comparable to freshly prepared micelle (Appendix Fig. 18). In summary, the biological safety and stability indicate PPCDQ nanoplatform has advantages in clinical translation.

## PPCDQ is a versatile nanovaccine platform against multiple pathogenic virus

In order to verify the generality of the PPCDQ platform, we assessed the immune efficacy of four additional nanovaccines (EBOV-RBD-NPs, MARV-GP1-NPs, NiV-Fc-G-NPs, and RABV-G-NPs). As shown in Fig. 7A,D,G,J, three of which target highly pathogenic pathogens for which no vaccine is currently available. The purity of each viral protein was first verified by SDS-PAGE, and the binding efficiency of PPCD micelles to each protein was

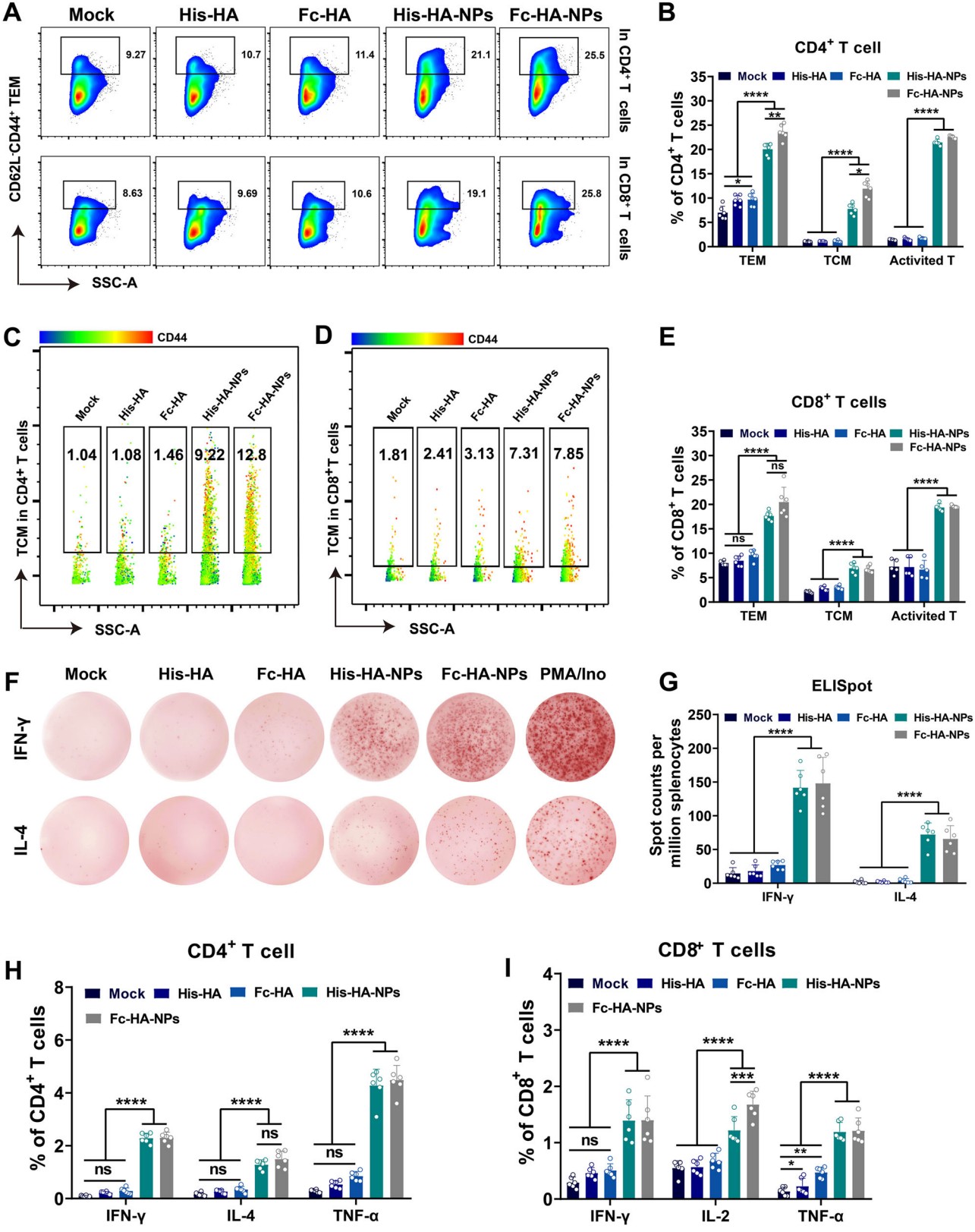

**Figure 4. T cell immune responses in HA@PPCDQ-vaccinated mice.**

(A) Representative flow cytometric plots of CD4$^+$ and CD8$^+$ effector memory T cells (TEM, CD62L$^-$CD44$^+$) at 2 weeks post boost vaccination. (B) Statistical results of TEM, TCM ($n = 6$ animals per group), and CD69$^+$ cells ($n = 5$ animals per group) in CD4$^+$ T cells. (C, D) Representative flow cytometric plots of central memory T cells (TCM, CD62L$^+$CD44$^+$) in CD4$^+$ T cells (C) and CD8$^+$ T cells (D). (E) Statistical results of TEM, TCM ($n = 6$ animals per group), and CD69$^+$ cells ($n = 5$ animals per group) in CD8$^+$ T cells. (F) ELISpot assays were conducted for IFN-γ (F, top) and IL-4 (F, bottom) secretion in splenocytes. (G) Statistical results of IFN-γ and IL-4 secreting cells ($n = 6$ animals per group). (H, I) Splenocytes were stimulated with homologous His-tagged HA for 18 h, Golgi Stop and Golgi Plug were added 6 h before the end of the stimulation. The percentages of IFN-γ$^+$, IL-4$^+$, and TNF-α$^+$ in CD4$^+$ T cells (H) and IFN-γ$^+$, IL-2$^+$, and TNF-α$^+$ in CD8$^+$ T cells (I) were determined by ICCS ($n = 6$ animals per group). Data information: Data in (B, E, G, H, I) are presented as mean ± SD. Statistical significance was calculated by one-way ANOVA with Tukey's multiple comparisons test. *$P < 0.05$, **$P < 0.01$, ***$P < 0.001$, ****$P < 0.0001$. Source data are available online for this figure.

evaluated using native PAGE (Appendix Fig. 19). The results of the pseudovirus neutralization test showed that the PPCDQ nanovaccines produced higher titers of neutralizing antibodies compared to the mixture of protein and AddaVax adjuvant (Fig. 7B,E,H). In particular, we found that all the mice produced virus-neutralizing antibodies (VNA) greater than 4 IU/mL (VNA titers >0.5 IU/mL are protective) in the second week after a boost immunization with RABV-G-NPs, while the mice in RABV-G + AddaVax group didn't generate detectable protective VNA (Fig. 7K). The data of ELISA assays showed that more antigen-specific total IgG was detected in PPCDQ nanovaccine groups (Fig. 7C,F,I,L). Furthermore, challenge protection experiments verified that RVG@PPCDQ can protect mice from lethal RABV challenges (Fig. 7M,N). Taken together, our results demonstrate that PPCDQ nanocarrier provides a universal platform which effectively enhance the humoral immune response of subunit vaccines to defend emerging and re-emerging pathogenic pathogens.

## Discussion

Limited immunogenicity of soluble proteins often restricts its usage in vaccine development. In this study, we designed a protein-polymer adjuvant system which can seamlessly display antigens on the surface and encapsulate adjuvants simultaneously, by utilizing the property of PLA-Porphyrin-Co$^{2+}$ that can effectively conjugate His-tagged proteins. Three different nanovaccines (including NPs His-HA-NPs, C-His-Fc-HA-NPs, and N-His-Fc-HA-NPs) all elicited robust humoral immunity and produced higher titers of HA-specific antibodies compared to traditional vaccine adjuvants. Further antigen trafficking experiments showed that HA@PPCDQ were more easily captured by APCs and prolonged the residence time of antigens in lymph nodes. Besides, we found that HA@PPCDQ promoted Th1 and Th2 response concomitantly and derived a higher frequency of antigen-specific CD8$^+$ TCM secreting IFN-γ, IL-2, and TNF-α, indicating that HA@PPCDQ was able to activate strong T cell immune responses. Furthermore, HA@PPCDQ also facilitated the antigen-specific GC B and GC Tfh cell generation to trigger a notable GC response.

The antibody response to protein epitope sites is limited (Cirelli et al, 2019). B cells that recognize prepotent epitope dominate the response, while the B cells that recognize other sites are at a disadvantage (i.e., immunodominance) (Angeletti et al, 2017; Angeletti et al, 2019; Angeletti and Yewdell, 2018; Sangesland and Lingwood, 2021). Compared to other nanoparticle vaccines, the conjugation strategy based on imidazole groups and transition metal ions Co$^{2+}$ does not need to introduce heterogeneous antigens,

thus better concentrating limited immunity on objective antigenic sites. Although several ferritin-based subunits have been reported to induce strong humoral immunity, a part of the immunity was still consumed to generate high titers of anti-ferritin antibodies (Jardine et al, 2013; Kanekiyo et al, 2013; Ma et al, 2020; Tokatlian et al, 2019). Previous studies have shown that slow delivery of antigens allowed GC B cells to encounter more "cryptic" epitopes and enhanced deposition of immune complexes on FDC (Cirelli et al, 2019). We found that HA@PPCDQ prolonged the residence time of antigen at the injection site and draining lymph nodes, while saponin adjuvant has been shown to be independent of the sustained release of antigen (Kensil, 1996; Pifferi et al, 2021), speculating that this "sustained release" is due to the carrier effect of PPCD nanoparticles. Early studies of antigen capture in lymph nodes revealed that most antigens are degraded in the medullary region, but a small amount reached lymphoid follicles and can be retained for long periods of time on follicular dendritic cells (FDCs) (Nossal et al, 1968; Phan et al, 2009; Phan et al, 2007; Tew et al, 1980). Our results showed more HA protein accumulated in B cell follicles and co-localized with FDCs and GC B cells after secondary immunization with HA@PPCDQ (Appendix Fig. 15), suggesting that the PPCDQ system facilitates cognate antigen encounters with GC B cells and deposition on FDCs. This may be one of the reasons for the higher affinity antibodies produced by HA@PPCDQ after immunization.

QS-21 is the most widely studied saponin adjuvant in recent years, due to its high purity, well solubility, and ability to elicit antibody responses and cellular immunity (Lacaille-Dubois, 2019). Like other saponin-containing liposomes (such as AS01, Matrix M, or other ISCOMs), the PPCDQ encapsulating QS-21 significantly increased the immunogenicity of IAV, EBOV, MARV, NiV, and RABV subunit vaccines, and showed good safety in mouse experiments. Previous studies have shown that fluorescently labeled QS-21 primarily targets CD169$^+$ macrophages in draining lymph nodes, induces caspase-1 activation and HMGB1 release, and coordinates the recruitment of innate immune cells and the activation of dendritic cells (Detienne et al, 2016). We found the frequencies of mature DCs in iLNs and neutrophils in spleens were significantly increased after HA@PPCDQ immunization, while the number of CD169$^+$ macrophages was greatly reduced. This is in agreement with previously published data showing that liposomes containing saponin can deplete the SSMs subpopulation (Silva et al, 2021). Nikolaos et al found that lymph node macrophages (SSMs and MSMs) undergo inflammasome-independent necroptosis after inactivated influenza virus vaccination, and confirmed that sialic acid (SA)-HA interaction and subsequent viral internalization are required for the induction of SSM death (Chatziandreou et al,

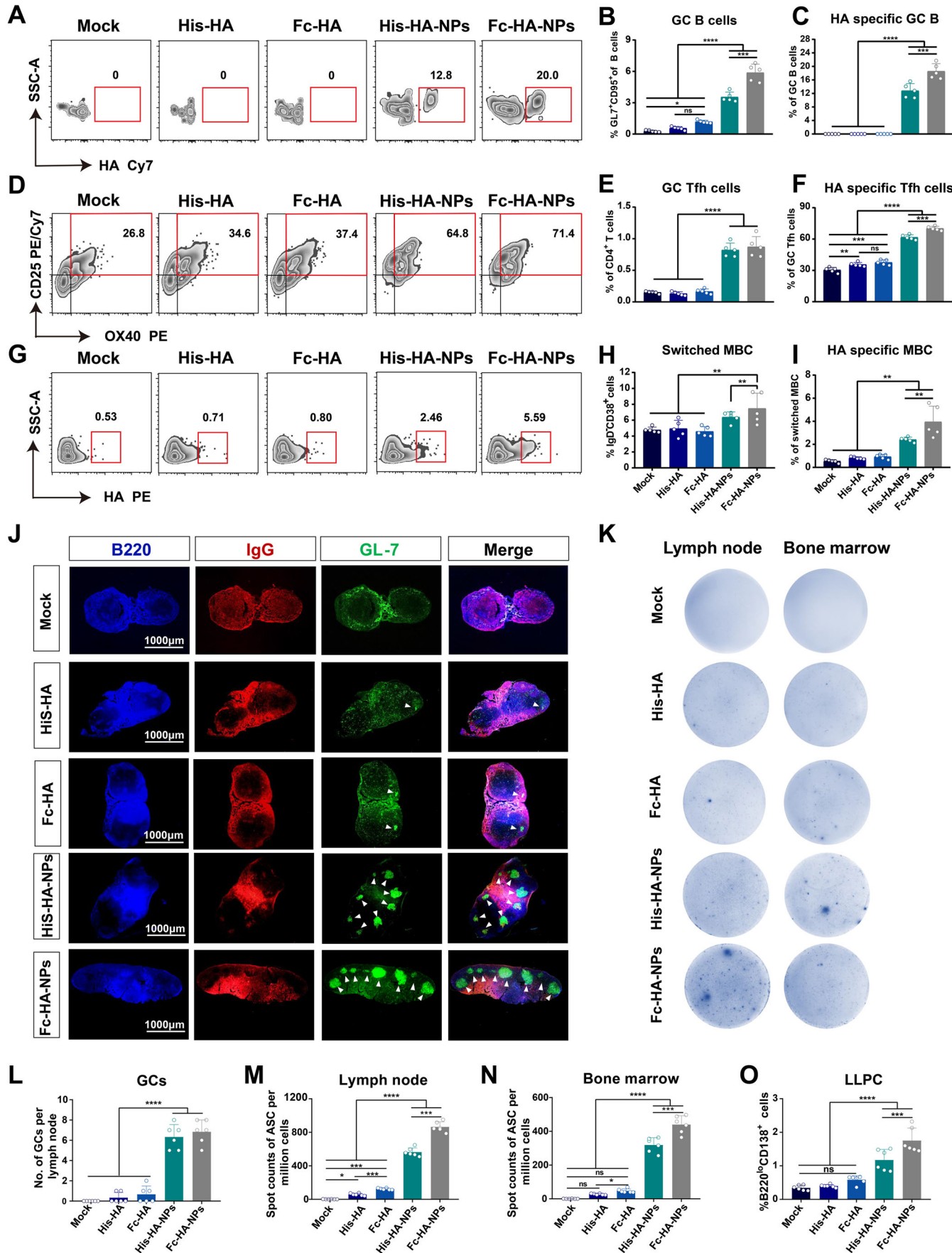

**Figure 5. HA@PPCDQ induces germinal center responses.**

(A) Representative flow cytometric plots of HA-specific GC B cells (CD95$^+$GL7$^+$HA$^+$). (B, C) Statistical results of GC B cells (CD95$^+$GL7$^+$) and HA-specific GC B cells (n = 5 animals per group). (D) The percentages of HA-specific GC Tfh cells (CD25$^+$OX40$^+$) were determined by AIM assay. (E, F) Statistical results of GC Tfh cells (CXCR5$^+$PD1$^+$) and HA-specific GC Tfh cells (n = 5 animals per group). (G) Representative flow cytometric plots of HA-specific switched memory B cells (MBC, IgD$^-$CD38$^+$HA$^+$). (H, I) Statistical results of switched MBCs (B220$^+$IgD$^-$CD38$^+$) and HA-specific switched MBC (n = 5 animals per group). (J) Cryosections of inguinal lymph nodes were incubated with a germinal center staining cocktail containing the following antibodies: Brilliant Violet 421™ anti-mouse/human CD45R/B220, Alexa Fluor® 594 Goat anti-mouse IgG and Alexa Fluor® 488 anti-mouse/human GL7. Scalar bar:1000 μm. (K) HA-specific ASCs were analyzed by ELISpot. (L) The number of GCs calculated according to the quantity of GL7 positive cell clusters (n = 6 animals per group). (M, N) The number of ASCs in lymph node (M) and bone marrow (N) (n = 6 animals per group). (O) Statistical results of LLPCs (B220$^{lo}$CD138$^+$) in bone marrow (n = 6 animals per group). Data information: Data presented in (B, C, E, F, H, I, L–O) are mean ± SD. Adjusted P values were calculated by one-way ANOVA with Tukey's multiple comparisons test. *P < 0.05, **P < 0.01, ***P < 0.001, ****P < 0.0001. Source data are available online for this figure.

2017). Our data showed that immunization with HA or Fc-HA protein alone did not cause changes in the number of SSM, whereas immunization with His-HA + QS-21 and His-HA-NP results in the same degree of SSM depletion, suggesting SSM depletion is caused by inflammation-induced by QS-21 loaded in the HA@PPCDQ system. The destruction of SSMs is a common feature associated with inflammation and viral or bacterial infection, while QS-21 has been verified to activate ASC-NLRP3 inflammasome and subsequent IL-1β/IL-18 release (Gaya et al, 2015; Marty-Roix et al, 2016). It has also been found that the destruction of SSM by inflammation impairs B cell responses to secondary infection (Gaya et al, 2015). However, the antibody response is greatly enhanced after the second immunization in our current study. Although SSM is still in a state of depletion after the second immunization. Combined with previous studies, it is speculated that the drainage of QS-21 to the subcapsular sinus may lead to the activation of inflammasome in SSMs and lead to the death of SSMs. This death event can further transmit inflammatory signals and recruit immune cells to mobilize innate and adaptive immunity (Garcia et al, 2012; Kastenmüller et al, 2012; Sagoo et al, 2016). This may be another important reason why HA@PPCDQ elicits strong humoral and cellular immunity. Additionally, QS-21-induced NLRP3 inflammasome activation may be responsible for the recruitment of neutrophils to the spleen after HA@PPCDQ immunization. Because circulating neutrophils are rapidly recruited to sites of inflammation in response to infectious and inflammatory stimuli (Kastenmüller et al, 2012; Lok and Clatworthy, 2021). Meanwhile, CD169$^+$ macrophages, Caspase-1 and MyD88 deficiencies have been shown to affect the recruitment of neutrophils to lymph nodes after QS-21 immunization (Detienne et al, 2016). The recruited neutrophils may regulate the subsequent adaptive immunity induced by HA@PPCDQ in many aspects, such as by interacting with DCs for antigen presentation (Bennouna and Denkers, 2005; Lok and Clatworthy, 2021), activating B cells (Balázs et al, 2002; Kolaczkowska and Kubes, 2013), promoting T cell immune response (Kesteman et al, 2008; Tillack et al, 2012), etc.

The prerequisite for subunit vaccines to work is that antigens can be drained to lymph nodes, taken up by DC cells, processed, and presented to T cells (Hong et al, 2020). The results of antigen tracing showed that HA@PPCDQ could drain to lymph nodes of the whole body rapidly and co-localize with DCs in the medullary region. Moreover, HA@PPCDQ were more easily taken up by DCs and macrophages. This suggests that PPCDQ nanocarriers promote antigen internalization by APCs, which is consistent with the notion that nanoparticles are more easily captured by APCs and stimulate their maturation (Feng et al, 2019; Su et al, 2023; Wang

et al, 2018). Efficient activation of naive T cells by DCs leads to their clonal expansion and differentiation into effector T cells and memory T cells (Bousso, 2008). We found that mice immunized with HA@PPCDQ generated a great number of antigen-specific CD4$^+$ TCMs and TEMs, illustrating that CD4$^+$ T cells were further activated by mature DCs. Previous studies have shown that saponin adjuvants can enhance CD8$^+$ T cell responses in mice by promoting antigen cross-presentation of DCs (den Brok et al, 2016; Newman et al, 1992; Schnurr et al, 2009). Significantly increased antigen-specific CD8$^+$ T cells after immunization indicated that HA@PPCDQ could promote DC cross-presentation of HA and efficient activation of CTLs (Lacaille-Dubois, 2019; Marty-Roix et al, 2016; Welsby et al, 2016).

The germinal center is an important site for B cell to undergo somatic hypermutation (SHM) and produce high-affinity antibodies (Bryant and Hodgkin, 2017). Our data showed that the number of GCs produced by HA@PPCDQ was about 8 times more than those of pure protein vaccine after 1 week of booster immunization. This would indicate that HA@PPCDQ can improve the quantity and quality of subunit vaccine-induced germinal centers. B cells shuttle between the light and dark areas of the germinal center, centrocytes (B cells in the light area) acquire and present material from the FDC to Tfh cells, and receive stimulatory signals that cause them to re-enter the dark area for further proliferation and mutation (Mayer et al, 2017). We found that HA@PPCDQ induced a higher frequency of antigen-specific Tfh and GC B cells, suggesting that the production of high-titer HA antibodies benefits from the cooperation between Tfh and GC B cells. This finding was consistent with the early report that saponins can induce antigen entry into relevant B cells and improve Tfh cell quality (Silva et al, 2021). HA-specific MBCs can quickly respond when influenza virus infects, further differentiate into effector B cells to enter the germinal center, or directly differentiate into plasma cells to produce anti-HA antibodies. Higher percentages of MBCs and LLPCs in the HA@PPCDQ immunized group provided long-term protection to against IAV infection.

The Fc-fusion protein can not only exert the biological activity of the fused protein, but also prolong its plasma half-life and specifically bind to the Fc receptor (FcRn) in the body (Czajkowsky et al, 2012). Previous research has shown that the Fc domain enhanced the immunogenicity of recombinant HA1 protein and elicits overt humoral immune response and local mucosal IgA antibodies (Ochsner et al, 2021; Yu et al, 2015). In this study, a unique feature of our vaccine design was the further display of Fc-HA dimers in addition to HA monomers. In order to exert the efficacy of the Fc domain, we added a His tag to the N-terminus of

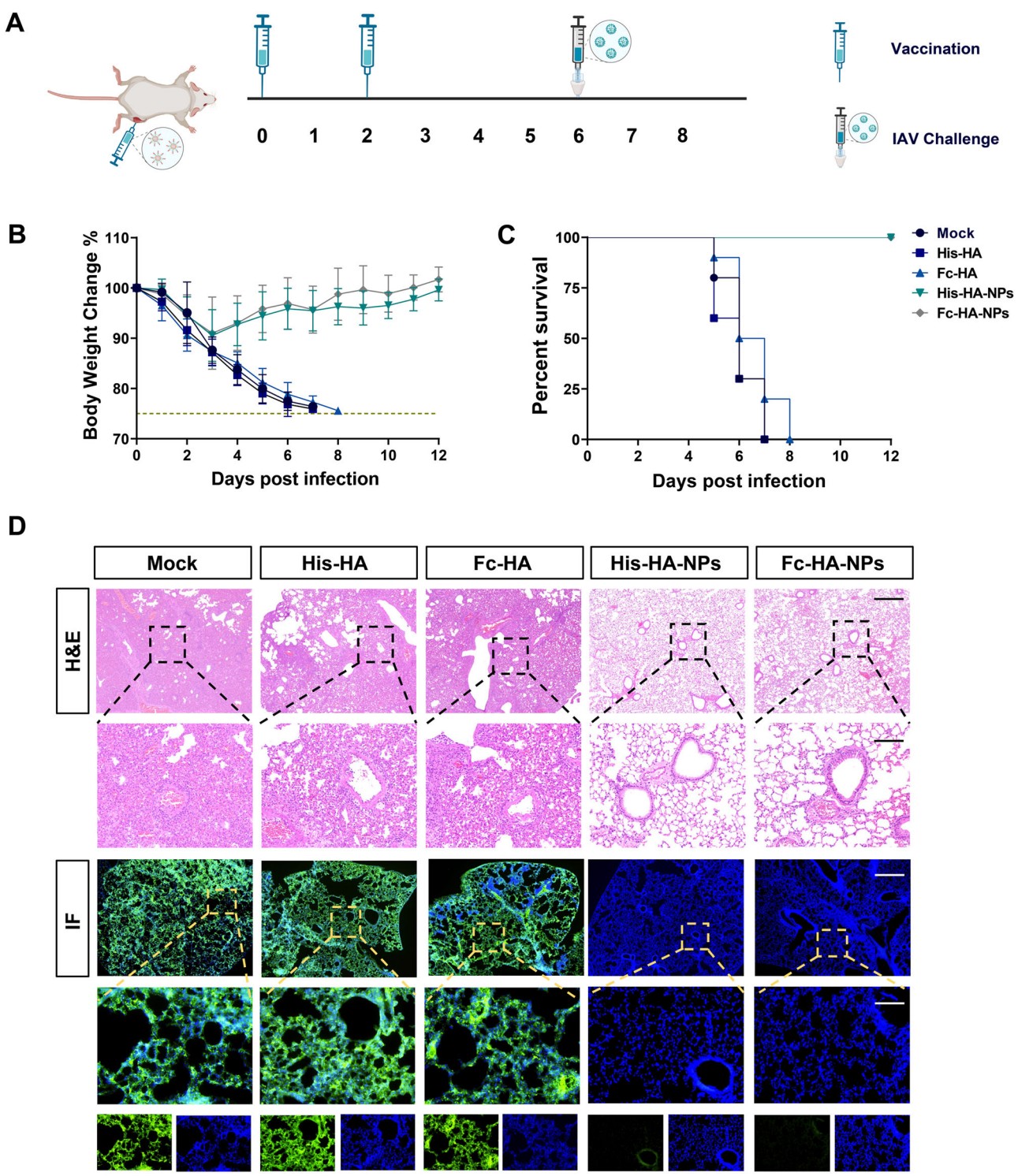

**Figure 6. HA@PPCDQ protects mice from lethal IAV challenge.**

(**A**) Schematic of C57BL/6 mice vaccination and challenge. Mice within each group were prime/boost-vaccinated with different vaccines (20 μg/mouse) at week 0 and week 2. Four weeks post boost, mice were challenged with authentic IAV. All mice were euthanized 2 weeks post challenge. (**B**) The body weight changes of mice. ($n = 10$ animals per group). Mice losing 25% of their pre-challenge body weight were euthanized. (**C**) Survival rates were monitored for 12 dpi ($n = 10$ animals per group). (**D**) H&E staining and IFA against IAV-NP protein were evaluated in the lungs of each mouse. Scale bar: 100 μm. Data information: Data presented in (**B**, **C**) are mean ± SD. Statistical significance in (**C**) was calculated by log-rank (Mantel-Cox) test. The experiments shown are representative of at least two independent experiments. Source data are available online for this figure.

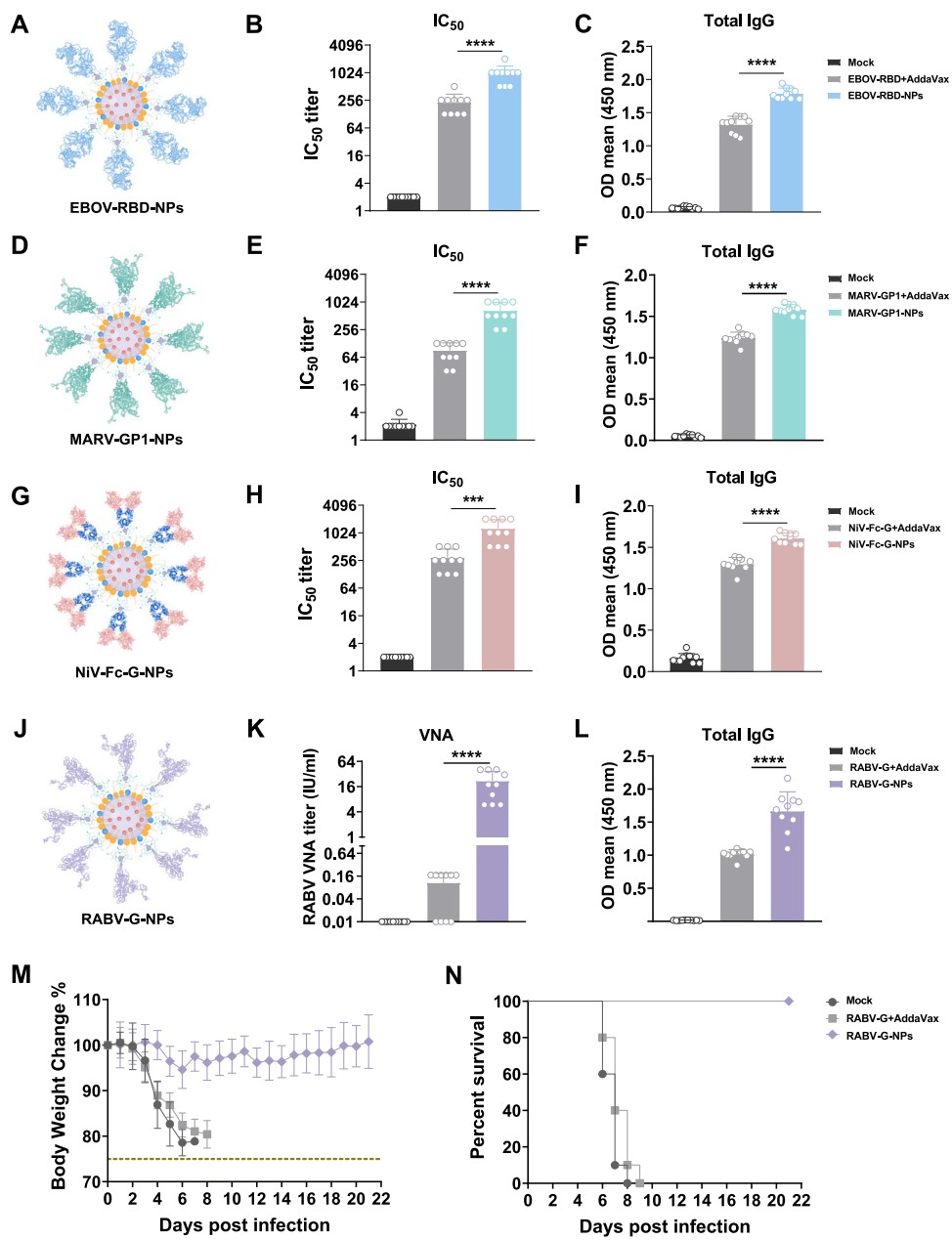

**Figure 7. PPCDQ is a universal nanovaccine platform against multiple pathogenic viruses.**

(A) Schematic illustration of EBOV-RBD-NPs. (B, C) Neutralizing antibody titers against rVSV-ΔG-EBOV/GP-GFP (B) and EBOV-RBD-specific total IgG (C). (n = 10 animals per group). (D) Schematic illustration of MARV-GP1-NPs. (E, F) Neutralizing antibody titers against rVSV-ΔG-MARV/GP-GFP (E) and MARV-GP1-specific total IgG (F). (n = 10 animals per group). (G) Schematic illustration of NiV-Fc-G-NPs. (H, I) Neutralizing antibody titers against rVSV-ΔG-NiV/F + G-GFP (H) and NiV-G-specific total IgG (C). (n = 10 animals per group). (J) Schematic illustration of RABV-G-NPs. (K, L) Neutralizing antibody titers against CVS-B2c (K) and RABV-G-specific total IgG (L) (n = 10 animals per group). (M) Mean body weight loss was monitored for 21 days (n = 10 animals per group). Mice losing >25% of their pre-challenge body weight were euthanized. (N) Survival rates were monitored for 21 dpi (n = 10 animals per group). Data information: Data in (B, C, E, F, H, I, K, L) are mean ± SD and statistical significance was calculated by one-way ANOVA with Tukey's multiple comparisons test. ***P < 0.001, ****P < 0.0001. statistical significance in (N) was calculated by log-rank (Mantel-Cox) test. The experiments shown are representative of at least two independent experiments. Source data are available online for this figure.

HA to expose the Fc domain on NPs. Besides, we set the control group with the His tag added to the C-terminal of HA-Fc. However, the N-His-HA-Fc NPs did not show an excellent immunization effect, especially since the antibody growth was slow in the first 2 weeks after primary immunization. It may be due to the combination of Fc and FcRn, causing a part of N-His-HA-Fc

NPs to escape from endosomes and re-enter the blood circulation without being presented by DCs (Liu et al, 2015). Although C-His-Fc-HA-NPs elicited comparable antibody levels compared to His-HA-NPs at the dose of 20 μg (there was only a significant difference in the third week), C-His-Fc-HA-NPs induced more Antigen-specific CD4+ TCM and TEM, and a stronger B cell response. In the

low-dose group (1 μg), C-His-Fc-HA-NPs showed better immune effects. In addition, NiV-G fused with the Fc domain also induced high titers of neutralizing antibodies in the PPCDQ system. More importantly, the Fc-fusion protein can be easily captured by the Protein A/G affinity column, which simplifies the purification steps of the fusion protein and is more conducive to subsequent production and clinical transformation.

In conclusion, we propose a protein self-assembled vaccine platform based on PPCD nanocarrier and QS-21 immunostimulant, which enhances antigen uptake by APCs and elicits robust T cell and B cell responses for protection against lethal virus infection (Fig. EV1). The synergy between QS-21 and PPCD triggers a series of subsequent cascade immune responses by activating APC and producing cytokines, greatly improving the immunogenicity of the pure protein. PPCD micelles integrated the antigen and adjuvant as a whole, which increased the stability of the nanovaccine, and its antigen display further facilitated the immunogenicity of the protein. In addition, this strategy has the characteristics of "self-assembling", rapid preparation, bio-safe, stable process, and can freely match the key antigens of various pathogens, especially for rapidly mutating envelope virus. Experimental results with multiple viral antigens indicate that PPCDQ is a promising universal protein-polymer adjuvant system, which can effectively enhance the cellular and humoral immune response for subunit vaccines against pathogenic pathogens.

# Methods

### Reagents and tools table

| Reagent/Resource | Reference or source | IDENTIFIER or catalog number |
|---|---|---|
| **Experimental models** | | |
| MDCK cells (*C. familiaris*) | ATCC | Stock No#CCL-34 |
| RAW264.7 cells (*M. musculus*) | ATCC | Stock No#TIB-71 |
| HEK-293T cells (*H. sapiens*) | ATCC | Stock No#CRL-11268 |
| BHK-21 cells (*M. musculus*) | ATCC | Stock No#CCL-10 |
| DC2.4 cells (*M. musculus*) | Feng hui Biotechnology | Stock No#CL0083 |
| Expi 293F cells (*H. sapiens*) | Shi et al, (2022) (Journal of Virology) | Prof. Dr. Guiqing Peng, HZAU, China |
| CD11c-EYFP (*M. musculus*) | Jackson Lab | Stock No#008829 |
| C57BL/6 (*M. musculus*) | Jackson Lab | Stock No#000664 |
| ICR (*M. musculus*) | Jackson Lab | Stock No#009122 |
| **Recombinant DNA** | | |
| pCMV3-EBOV-GP-C-Flag | Sino Biological | Cat. No# VG40304-CF |
| pCAGGS-C-His-HA | This study | N/A |
| pCAGGS-C-His-Fc-HA | This study | N/A |
| pCAGGS-N-His-Fc-HA | This study | N/A |
| pCAGGS-C-Fc-HA | This study | N/A |
| pCAGGS-EBOV-RBD | This study | N/A |
| pCAGGS-MARV-GP1 | This study | N/A |

| Reagent/Resource | Reference or source | IDENTIFIER or catalog number |
|---|---|---|
| pCAGGS-NiV-G | This study | N/A |
| pCDNA3.1(+)-NiV-F | This study | N/A |
| pCDNA3.1(+)-NiV-G | This study | N/A |
| pCDNA3.1(+)-MARV-GP | This study | N/A |
| pCDNA3.1(+)-EBOV-GP | This study | N/A |
| **Antibodies** | | |
| PerCP/Cyanine5.5 anti-mouse CD45R/B220 | BD Biosciences | Cat. No#552771 |
| PE anti-mouse CD86 | Biolegend | Cat. No#105008 |
| APC anti-mouse CD80 | Biolegend | Cat. No#104714 |
| PE/Cyanine7 anti-mouse I-A/I-E | Biolegend | Cat. No#107630 |
| FITC anti-mouse CD11c | Biolegend | Cat. No#117306 |
| FITC anti-mouse CD169 (Siglec-1) | Biolegend | Cat. No# 142406 |
| PE anti-mouse F4/80 | Biolegend | Cat. No#123110 |
| APC/Cyanine7 anti-mouse/human CD11b | Biolegend | Cat. No#101226 |
| FITC anti-mouse CD4 | Biolegend | Cat. No#100406 |
| APC/Cyanine7 anti-mouse CD3 | Biolegend | Cat. No#100222 |
| Pacific Blue™ anti-mouse CD8a | Biolegend | Cat. No#100725 |
| PE/Cyanine7 anti-mouse CD44 | Biolegend | Cat. No#103030 |
| PerCP/Cyanine5.5 anti-mouse CD62L | Biolegend | Cat. No#104432 |
| APC/Fire™ 750 anti-mouse CD69 | Biolegend | Cat. No#104549 |
| FITC anti-mouse/human CD45R/B220 | Biolegend | Cat. No#103206 |
| Alexa Fluor® 647 anti-mouse/human GL7 | Biolegend | Cat. No#144606 |
| PE anti-mouse CD95 (Fas) | Biolegend | Cat. No#152608 |
| PE anti-mouse CD279 (PD-1) | Biolegend | Cat. No#109104 |
| APC anti-mouse CD185 (CXCR5) | Biolegend | Cat. No#145506 |
| APC anti-mouse CD38 | Biolegend | Cat. No#102712 |
| FITC anti-mouse IgD | Biolegend | Cat. No#405703 |
| PE anti-mouse CD138 | Biolegend | Cat. No#142504 |
| PE anti-mouse IL-2 | Biolegend | Cat. No#503808 |
| APC anti-mouse TNF-α | Biolegend | Cat. No#506308 |
| Brilliant Violet 605™ anti-IFN-γ | Biolegend | Cat. No#505840 |
| PE anti-mouse IL-4 | Biolegend | Cat. No#504104 |
| Brilliant Violet 421™ anti-mouse CD279 (PD-1) | Biolegend | Cat. No#135221 |
| PE anti-mouse CD134 (OX-40) | Biolegend | Cat. No#119409 |
| PE/Cyanine7 anti-mouse CD25 | Biolegend | Cat. No#101916 |
| Brilliant Violet 421™ anti-mouse CD45R/B220 | Biolegend | Cat. No#103240 |
| Alexa Fluor® 594 Goat anti-mouse IgG | Biolegend | Cat. No#405326 |
| Alexa Fluor® 488 anti-mouse/human GL7 | Biolegend | Cat. No#144611 |

| Reagent/Resource | Reference or source | IDENTIFIER or catalog number |
|---|---|---|
| BV510 anti-mouse CD35 | BD Biosciences | Cat. No#740132 |
| Alexa Fluor™ 488 Goat anti-Mouse IgG (H + L) | Invitrogen | Cat. No#A-11001 |
| Anti-NP (Influenza A) | Immune technology | Cat. No#IT-003-023M1 |
| HRP-goat anti-mouse IgG(H + L) | Biodragon | Cat. No#BF03001 |
| HRP-goat anti-mouse IgG1 | Biodragon | Cat. No#BF03002 |
| HRP-goat anti-mouse IgG2a | Biodragon | Cat. No#BF03003 |
| HRP-goat anti-mouse IgG2b | Biodragon | Cat. No#BF03004 |
| **Oligonucleotides and sequence-based reagents** | | |
| PCR primers | This study | Appendix Table S1 |
| **Chemicals, enzymes and other reagents** | | |
| L-Lactide | Macklin | Cat. No#4511-42-6 |
| D (+) -Lactide | Macklin | Cat. No#13076-17-0 |
| Tin (II) 2-Ethylhexanoate | Macklin | Cat. No#301-10-0 |
| N-(3-Dimethylaminopropyl)-N′-ethylcarbodiimide hydrochloride | Macklin | Cat. No#25952-53-8 |
| 4-Dimethylaminopyridine | Macklin | Cat. No#1122-58-3 |
| Cobaltous chloride | Macklin | Cat. No#7646-79-9 |
| DSPE-PEG2000 | Yuanye Bio-Technology | Cat. No#147867-65-0 |
| Quillaja Saponaria-21 | Yuanye Bio-Technology | Cat. No#141256-04-4 |
| Alum | InvivoGen | Cat. No#vac-alu-250 |
| AddaVax | InvivoGen | Cat. No#10adx-vac |
| AddaS03 | InvivoGen | Cat. No#vac-as03-10 |
| Receptor-disrupting enzyme (RDE) | Sigma-Aldrich | Cat. No#C8772 |
| TPCK treated trypsin (T1426) | Sigma-Aldrich | Cat. No#9002-07-7 |
| Recombinant Rabies virus Glycoprotein | AnyGo Technology | Cat. No#AK100H1 |
| Recombinant Murine IL-4 | PEPROTECH | Cat. No#214-14 |
| Recombinant Murine GM-CSF | PEPROTECH | Cat. No#315-03 |
| IFluor™ 594 succinimidyl ester | AAT Bioquest | Cat. No#1029 |
| IFluor™ 488 succinimidyl ester | AAT Bioquest | Cat. No#1023 |
| Cyanine-5 monosuccinimidyl ester | AAT Bioquest | Cat. No#151 |
| Cyanine7 monosuccinimidyl ester | AAT Bioquest | Cat. No#161 |
| FluoroFix™ Buffer | Biolegend | Cat. No#422101 |
| Intracellular Staining Perm Wash Buffer (10X) | Biolegend | Cat. No#421002 |
| PE Streptavidin | Biolegend | Cat. No#405203 |
| Native Gel Sample Loading Buffer | Beyotime | Cat. No#P0016N |
| Coomassie Blue Super-Fast Staining Solution | Beyotime | Cat. No#P0017F |

| Reagent/Resource | Reference or source | IDENTIFIER or catalog number |
|---|---|---|
| 5× TBE Running Buffer | Monad | Cat. No#CR00401S |
| GolgiPlug™ | BD Biosciences | Cat. No#555029 |
| GolgiStop™ | BD Biosciences | Cat. No#554724 |
| Zombie Aqua™ Fixable Viability Kit | Biolegend | Cat. No# 423101 |
| Mouse IL-4 Precoated ELISPOT Kit | DAKEWE | Cat. No#2210402 |
| Mouse IFN-γ Precoated ELISPOT Kit | DAKEWE | Cat. No#2110002 |
| Biotinylation Kit | Genemore | Cat. No#G-MM-IGT |
| **Software** | | |
| FlowJo v10 | | https://www.flowjo.com/solutions/flowjo/ |
| Prism v8 | | https://www.graphpad.com/ |
| Illustrator v27.7 | | https://www.adobe.com/products/illustrator.html |
| ImageJ | | https://imagej.nih.gov/ij/index.html |
| NIS-Elements Viewer v5.21.00 | | https://www.microscope.healthcare.nikon.com/products/software/nis-elements/viewer |
| Living Image v4.4 | | https://www.revvity.co.jp/software-downloads/in-vivo-imaging |
| Harmony v4.9 | | https://www.perkinelmer.com.cn/lab-products-and-services/cellular-imaging/harmony-video-tutorials.html |
| Arivis Vision4D v4.0 | | https://www.arivis.com/arivis-news/arivisvision4d40 |
| **Other** | | |
| Opera Phenix High-Content Screening System | PerkinElmer | |
| IVIS Spectrum | PerkinElmer | |
| Cytoflex LX/ Cytoflex | Beckman Coulter | |
| Nikon A1HD25 confocal microscopy | Nikon | |
| Light Sheet Fluorescence Microscopy (LSFM, Zeiss Z.1) | Zeiss | |
| BX63 fluorescence microscope | Olympus | |

## Ethics statements

All animals involved in this study were raised in the Animal Facility at Huazhong Agricultural University. All animal experiments performed were grouped by random procedure and strictly following the recommendations in the Guide for the Care and Use of Laboratory Animals of the Ministry of Science and Technology of the People's Republic of China. All experimental procedures were approved by the Scientific Ethics Committee of Huazhong Agricultural University (permit number: HZAUMO-2022-0152).

## Cell culture

MDCK cells (Madin-Darby canine kidney), DC2.4 cells, RAW264.7 (Mouse Mononuclear Macrophages Cells), HEK-293T, BHK-21(Baby Hamster Kidney cell), and BSR cells (BSR (a clone of BHK-21) were cultured in Dulbecco's modified Eagle's medium (DMEM, BioChannel Biotechnology Co., Ltd) containing 10% fetal bovine serum (FBS, QmSuero Biotech Co., Ltd.) and 1% penicillin/streptomycin (Biosharp). Expi 293F cells were cultured in 293 cell culture (Union Biotechnology Co., Ltd.).

## Virus strains

Influenza A/Puerto Rico/8/1934 (H1N1) virus (PR8) were kindly donated by Dr. Hongbo Zhou (Huazhong Agricultural University, Wuhan, China) (Zhu et al, 2020). Rabies challenge virus strain, CVS-24 and CVS-B2c, were passaged in BHK-21 cells and preserved in our laboratory. All experiments related to the virus were completed in the P2 laboratory.

## Plasmid construction, protein expression, and purification

Humanized HA ectodomain sequence (GenBank accession number NP_040980.1, residues 18–522, Tsingke), MARV-GP sequence (GenBank accession number AGL73411.1, residues 1 to 681, Genscript), NiV-F sequence (GenBank accession number AAY43915.1, residues 1 to 546, Genscript), and NiV-G sequence (GenBank accession number AAY43916.1, residues 1 to 602, Genscript) was synthesized to pCDNA3.1(+) vector and cloned into pCAGGS vector with a N-terminal IgG1 signal peptide, a C-terminal 8 × His tag or a Fc fragment of mouse IgG2a. EBOV-GP gene (subtype Zaire, strain Mayinga 1976) was purchased from Sino Biological (Cat. No# VG40304-CF), and the sequence of receptor binding domain (RBD, residues 1 to 308) was cloned into pCAGGS vector for soluble protein expression. Primers used for plasmid construction were presented in Appendix Table S1. The recombinant plasmids were transfected into Expi 293 F cells with polyetherimide (PEI, Yeasen Biotechnology Co., Ltd.) according to the manufacturer's instruction, and the cells were cultured for another four days at 37 °C and 8% $CO_2$. The cell supernatants were harvested and purified according to a protocol described previously (Shi et al, 2022). Briefly, the proteins tagged with His were purified through a series of chromatography steps, including the HisTrap HP column and Superdex 200 High-Performance column (Cytiva life sciences). The Fc-fusion proteins were purified using a Protein A + G column (Beyotime Biotechnology Co., Ltd.). All the proteins were concentrated and exchanged in phosphate-buffered saline (PBS) buffer and stored at −80 °C.

## Synthesis and characterization of PPCDQ-NPs with different proteins

To prepare poly(lactic acid) (PLA), 2.1180 g of L-lactide, D-lactide, and 0.31 mL of Tin (II) 2-Ethylhexanoate (0.02 g/mL) were added to 21.0 mL of dry toluene solution and reacted at 130 °C for 2 days. The reaction product was freeze-dried to obtain PLA. Next, 162.3 mg of PLA and 53.4 mg of pyropheophorbide-a were added to anhydrous dichloromethane solution, and 38.6 mg of N-(3-

Dimethylaminopropyl)-N'-ethylcarbodiimide hydrochloride and 49.8 mg of 4-Dimethylaminopyridine were added and reacted in ice for 24 h to produce PLA-Pyro. About 1.0 mL of saturated methanol solution of cobaltous chloride was added to 50 mL of dichloromethane solution of PLA-pyro and stirred for 12 h to generate PLA-Pyro-$Co^{2+}$. DSPE-PEG2000 and PLA-Pyro-$Co^{2+}$ were mixed in equal proportions in an ethanol solution, and the mixture was added dropwise to the PBS solution to produce PPCD micelles. Finally, PPCD micelles were mixed with QS-21 and different viral proteins (IAV-HA, EBOV-RBD, MARV-GP1, NiV-Fc-G, or RABV-G), respectively, and self-assembled for 12 h at 4 °C. The final products were purified by the size-exclusion chromatography and concentrated with a 0.22 μm filter (Millipore) to obtain HA-NPs, EBOV-RBD-NPs, MARV-GP1-NPs, NiV-Fc-G-NPs, and RABV-G-NPs. The size and zeta potential of HA@PPCDQ were measured on a Malvern Zetasizer Nano ZS apparatus (Malvern, Worcestershire, UK).

## Size-exclusion chromatography (SEC)

His-HA, C-His-Fc-HA, and N-His-Fc-HA were incubated with PPCD in PBS buffer, respectively. Twenty-four hours later, the His-HA-NPs, C-His-Fc-HA-NPs, and N-His-Fc-HA-NPs were proceeded to size-exclusion chromatography (SEC), and display the UV absorption at 280 nm (protein) and 430 nm (PPCD micelle).

## Transmission electron microscopy (TEM)

Purified HA@PPCDQ samples were diluted to 100 μg/ml in PBS, and 5 μL of each was placed on 200 copper grids for 45 s, respectively. The grids were washed twice with 5 μl of distilled water. Excess liquid was removed, and samples were air-dried. Grids were imaged on an HT7700 (Hitachi, Japan) microscope at 100.0 kV.

## Native PAGE analysis of proteins binding

To investigate the ability of PPCD micelles to load antigens, native PAGE was used to quantify the protein anchoring efficacy. In brief, PPCD micelles and C-His-HA protein were mixed at different ratios (2.5:1, 5:1, 10:1, 20:1, wt/wt), incubated at 4 °C for 6 h, then added Native Gel Sample Loading Buffer (Beyotime) for PAGE. And the equal amounts of C-His-HA protein was electrophoresed together as a positive control. Native PAGE separating gel was prepared containing 30% Acrylamide, 20% Glycine, 10% ammonium persulfate (APS), and 5× TBE buffer. A MonTrack 5× TBE Running Buffer (Monad) was used, and the samples were electrophoresed in ice water at 120 mV for 120 min. The gel was stained with Coomassie Blue Super-Fast Staining Solution (Beyotime) for 30 min and destained with ddH$_2$O with shaking at room temperature for 2 h. All gels were scanned by GelScanner 2100XL (DHS Life Science & Technology) and analyzed by ImageJ software for grayscale value to calculate binding efficiency. The data come from the results of three independent repeated experiments. The binding efficiency of micelles was calculated by the following equation:

$$\text{Binding}(\%) = [1 - V_{\text{sample}}/V_{\text{positive}}] \times 100$$

## Hemolysis assay in vitro

Hemolytic activity of His-HA-NPs, C-His-Fc-HA-NPs, and N-His-Fc-HA-NPs were tested by direct contact methods as previously described (Huang et al, 2013). Anticoagulated mouse blood (0.02 mL) was added to 1 mL of PBS containing different concentrations of HA@PPCDQ or PBS as a negative control or 1% Triton as a positive control. Then, the contents of the tubes were gently mixed and placed in an incubator at 37 °C. After incubation for 3 h, the samples were centrifuged at 1000 rpm for 10 min and absorbance of the supernatant of each tube was measured by SpectraMax 190 spectrophotometer (Molecular Devices, CA, USA) at 545 nm. The rate of hemolysis was calculated according to the following equation:

$$\text{Hemolysis}(\%) = \frac{\text{OD}_{\text{sample}} - \text{OD}_{\text{negative}}}{\text{OD}_{\text{positive}} - \text{OD}_{\text{negative}}} \times 100$$

## Mouse immunization and challenge test

For the IAV model, Mice were immunized with HA@PPCDQ at a dose of 1–20 μg (100 μL total in PBS buffer) or same mass of HA/HA-Fc with other adjuvants (Alum, AddaVax, or AddaS03) by hindlimb intramuscular (I.M.) injection. The challenge protection experiment was carried out 4 weeks after the booster immunization. Mice were intranasally challenged with $1.8 \times 10^4$ times the median tissue culture infectious dose ($\text{TCID}_{50}$) of PR8 virus for the Influenza A virus model and the weight of mice was monitored for 12 consecutive days.

For the EBOV, MARV, NiV, and RABV model, mice were intramuscularly immunized with EBOV-RBD-NPs, MARV-GP1-NPs, NiV-Fc-G-NPs, or RABV-G-NPs at a dose of 10 μg, respectively. Same mass of different viral proteins with AddaVax adjuvant were immunized as controls. All mice were immunized at week 0 and boosted with a second vaccination at week 2. Mice immunized with RABV-G-NPs were intracerebrally challenged with 50 LD50 CVS-24 at four weeks after boost vaccination. Then, clinical signs and body weight were observed for 21 consecutive days.

## Hemagglutination inhibition (HI) assay

The titers of HI in sera samples were tested by a standard HI assay as previously described (Si et al, 2022). Briefly, serum was diluted with receptor-disrupting enzyme (RDE, Sigma-Aldrich) at a volume ratio of 1:4 and incubated at 37 °C for 18–20 h. After inactivation at 56 °C for 30 min, twofold serial dilutions of RDE-inactivated sera were added to a V-well microtiter plate (25 μL/well) and incubated with 4 HA units of PR8 virus (25 μL/well) at room temperature (RT) for 30 min. Finally, 50 μL of 0.5% (v/v) chicken red blood cells (SenBeJia Biological Technology Co., Ltd.) in Alsever's Solution were added to each well and incubated for 30 min at RT. The plate was tilted to examine RBC teardrop formation in the well and HI titers were read as the highest dilution of sera samples that inhibited hemagglutination. Titers below the detection limit (<4) were assigned a value of 4 for statistical analysis.

## Microneutralization (MN) assay

The titers of anti-IAV neutralizing antibody were tested by an MN assay as previously described (Cuevas et al, 2022). MDCK cells were seeded into 96-well flat-bottom plates in DMEM supplemented with 10% FBS and 1% penicillin to produce a monolayer. RDE-inactivated sera samples were twofold serially diluted in DMEM and incubated with 100 TCID50 of virus for 1 h at 37 °C. Then, the mixture of sera and virus was applied to MDCK monolayers with 1 μg/mL TPCK (tolylsulfonyl phenylalanyl chloromethyl ketone)-treated trypsin (Sigma-Aldrich) and incubated for 1 h (37 °C, 5% $CO_2$) and then the medium was removed from each well and replaced with fresh DMEM with 1 μg/mL TPCK. The cell was cultured for another 3 days at 37 °C in 5% $CO_2$. Finally, the supernatants were used in a hemagglutination (HA) assay with red blood cells to determine the presence of the virus and the degree of neutralization. The MN titer titers were defined as the reciprocal of the highest dilution of the serum sample that cannot agglutinate red blood cells completely. Titers below the detection limit (<4) were assigned a value of 4 for statistical analysis.

## ELISA assays

ELISA assays were implemented to determine antigen-specific total IgG and antibody isotypes. For IAV, the recombinant HA protein was diluted with coating buffer and added to ELISA plates, which were coated overnight at 4 °C. ELISA plates were washed three times with phosphate-buffered saline (PBS)-Tween (PBST) and blocked with 5% low-fat milk at 37 °C for 2 h. The serum was diluted and incubated in ELISA plates at 37 °C for 2 h. ELISA plates were washed three times with PBST, and then horseradish peroxidase (HRP)-conjugated goat anti-mouse Abs (IgG (1:4000 dilution), IgG1 (1:2000 dilution), IgG2a (1:2000 dilution), and IgG2a (1:2000 dilution)) were added to wells at 37 °C for 1 h. Tetramethyl-benzidine (TMB) substrate was added at room temperature and incubated away from light for 30 min. Then, 2 M $H_2SO_4$ was added to terminate the reaction. For EBOV, MARV, NiV, and RABV, the coating antigen was replaced with EBOV-RBD, MARV-GP1, NiV-G, or RABV-G, other steps are similar to those for IAV. The value at OD450 nm was immediately determined by a SpectraMax 190 spectrophotometer (Molecular Devices, CA, USA).

## Virus-neutralizing antibodies (VNA) assay

The RABV-neutralizing antibodies titers in sera samples of RVG@PPCDQ-NPs immunized mice were quantified by fluorescent antibody virus neutralization (FAVN) assay as described previously (Zhang et al, 2020). Briefly, inactivated serum was threefold serially diluted in 96-well microplates starting at 1:9 in DMEM, and each sample was made in quadruplicate. For each experiment, a virus regression control and a standard serum control were set up. Then, the suspension of CVS-B2c was added to each well, and the 96-well plates were placed at 37 °C for 1 h. Subsequently, suspended cells of BSR were added to 96-well plates and cultured at 37 °C for 60 h. Cells were fixed with 80% ice acetone for 30 min and stained with FITC-conjugated antibodies against RABV phosphoprotein (1:1000 dilution, stored in our laboratory) at 37 °C for 1 h. Fluorescent spots were observed with a Zeiss

fluorescence microscope. VNA titers were calculated based on the result of the reference serum (WHO) and measured in international units per milliliter (IU/mL). Titers below the detection limit (<0.17) were assigned a value of 0.01 for statistical analysis.

## Generation of pseudotyped recombinant vesicular stomatitis virus (rVSV)

Recombinant VSV containing genomic inserts for expression of green fluorescent protein (GFP, rVSV-ΔG-GFP) was kindly provided by Dr. Guiqing Peng (Huazhong Agricultural University, Wuhan, China). rVSV-ΔG-EBOV/GP-GFP, rVSV-ΔG-MARV/GP-GFP, rVSV-ΔG-NiV/F + G-GFP were prepared as described previously (Doyle et al, 2021; Milligan et al, 2022; Wang et al, 2022; Whitt, 2010). Briefly, pCMV3-EBOV-G and pCDNA3.1(+)-MARV-GP plasmids were transfected into HEK-293T cells using jetPRIME transfection reagent (Polyplus), respectively. pCDNA3.1(+)-NiV-F and pCDNA3.1(+)-NiV-G plasmids were co-transfected together. After 24 h, the transfected cells were infected with rVSV-ΔG-GFP at a multiplicity of infection (MOI) of 1 for 2 h, and then the cell supernatant was removed and replaced with a fresh medium. Cell supernatants were harvested and centrifuged after 48 h of culture and stored at −80 °C. The resulting rVSV pseudotyped viruses were titrated on BHK-21 cell monolayers and calculated using the Reed-Muench method. All experiments related to pseudovirus were performed in the P2 laboratory.

## Pseudotyped-virus neutralization assay

Black-walled 96-well plates (LABSELLECT) were seeded with $2 \times 10^4$ BHK-21 cells per well and incubated overnight. Inactivated mouse sera were individually diluted twofold in DMEM and mixed with the corresponding rVSV (rVSV-ΔG-EBOV/GP-GFP, rVSV-ΔG-MARV/GP-GFP, rVSV-ΔG-NiV/F + G-GFP; 100 TCID50/well) and incubated for 2 h at 37 °C. Following incubation, 100 μL rVSV/sera mixtures were added to cell monolayers and incubated for 36 h at 37 °C. Then, cells were fixed with 100 μL per well of 4% formalin for 20 min at room temperature. Fluorescent foci were counted using the Opera Phenix High-Content Screening System (PerkinElmer, Massachusetts, USA) and Harmony 4.9 software. Neutralization protection rate (50% inhibitory concentration, $IC_{50}$) was calculated by normalizing counts to an rVSV-only control.

## Flow cytometry (FCM)

To identify DCs and macrophages, inguinal lymph nodes or spleen were harvested and homogenized into single-cell suspensions using a syringe plunger and passed through a 40 μm nylon filter (SPL Life Sciences Co., Ltd). Red blood cells were lysed with ACK lysis buffer (BioSource, International, Inc., CA, USA). After washing two times, single-cell suspensions containing $10^6$ cells were blocked in PBS buffer with 0.2% bovine serum albumin (BSA). Next, cells were stained separately with cocktails of the following fluorescently labeled antibodies: PerCP/Cyanine5.5 anti-mouse CD45R/B220 (1:1000 dilution), PE anti-mouse CD86 (1:20 dilution), APC anti-mouse CD80 (1:20 dilution), PE/Cyanine7 anti-mouse I-A/I-E (1:300 dilution), FITC anti-mouse CD11c (1:300 dilution), FITC anti-mouse CD169 (Siglec-1) (1:100 dilution), PE anti-mouse F4/80

(1:20 dilution), APC/Cyanine7 anti-mouse/human CD11b (1:80 dilution). Cells were washed three times and then resuspended in PBS for acquisition. The gating strategies were described in Appendix Fig. S6.

To identify TCMs, TEMs, and CD69 positive cells in CD4$^+$ T cells or CD8$^+$ T cells, freshly isolated splenocytes were stained with cocktails of T cell surface markers for 30 min. The Zombie Aqua™ Fixable Viability Kit were used to gate live cells. The following indicated antibodies were used: FITC anti-mouse CD4 (1:200 dilution), APC/Cyanine7 anti-mouse CD3 (1:80 dilution), Pacific Blue™ anti-mouse CD8a (1:200 dilution), PE/Cyanine7 anti-mouse CD44 (1:80 dilution), PerCP/Cyanine5.5 anti-mouse CD62L (1:80 dilution), APC/Fire™ 750 anti-mouse CD69 (1:80 dilution). Cells were washed three times and then resuspended in PBS for FCM analysis. The gating strategies for splenocytes were described in Appendix Figs. S7A, S10A.

To identify total GC B cells, Tfh cells, MBCs, and LLPCs, freshly isolated LN or BM immune cell suspensions were stained with the surface cocktail containing the following antibodies, FITC anti-mouse/human CD45R/B220 (1:100 dilution), Alexa Fluor® 647 anti-mouse/human GL7(1:100 dilution), PE anti-mouse CD95 (Fas) (1:160 dilution), APC/Cyanine7 anti-mouse CD3(1:160 dilution), FITC anti-mouse CD4 (1:200 dilution), PE anti-mouse CD279 (PD-1) (1:20 dilution), APC anti-mouse CD185 (CXCR5) (1:80 dilution), PerCP/Cyanine5.5 anti-mouse CD45R/B220 (1:1000 dilution), APC anti-mouse CD38 (1:80 dilution), FITC anti-mouse IgD (1:200 dilution), PE anti-mouse CD138 (1:80 dilution). For HA-specific GC B cells quantitation, recombinant His-HA protein was conjugated to Cyanine7 (Cy7) by using Cy7 monosuccinimidyl ester according to the manufacturer's instructions. Cells were incubated with 10 μg/mL His-HA-Cy7 for 30 min prior to FCM analysis.

For HA-specific MBCs quantitation, recombinant His-HA protein was biotinylated by using Biotinylation Kit. single-cell suspensions were incubated with 5 μg/mL His-HA-Biotin for 30 min and washed. Cells then incubated with PE Streptavidin for 30 min prior to FCM analysis. The gating strategies were described in Appendix Figs. S13, S14.

Cell sorting was performed with a Cytoflex LX/ Cytoflex flow cytometer (Beckman Coulter) and data were analyzed using FlowJo software V_10.

## Intracellular cytokine staining (ICCS)

In order to identify antigen-specific T cells, freshly isolated splenocyte suspensions ($8 \times 10^6$/1000 μL in a 12-well plate) were stimulated with 10% RPMI (negative control, Gibco) or 10–30 μg/mL homologous His-tagged HA for 18 h (37 °C, 5% $CO_2$). PMA/ionomycin (DAKEWE) was used as a positive control. Golgi Stop and Golgi Plug (BD Biosciences) were added 6 h before the end of the stimulation according to the manufacturer's instructions. Cells were washed two times with PBS and blocked in PBS with 0.2% BSA. The Zombie Aqua™ Fixable Viability Kit was used to exclude dead cells. Samples were incubated for an additional 30 min with the surface cocktail as described above for TCMs and TEMs acquisition. After washing two times with PBS, cells were performed with a FluoroFix™ Buffer for 20 min at RT in the dark and permeabilized with intracellular staining perm wash buffer. Subsequently, cells were incubated for intracellular staining with

the following antibodies (30 min, 4 °C): PE anti-mouse IL-2 (1:80 dilution), APC anti-mouse TNF-α (1:80 dilution), Brilliant Violet 605™ anti-IFN-γ (1:80 dilution), PE anti-mouse IL-4 (1:80 dilution). The percentages of cytokine-specific T cells were analyzed by flow cytometry. The gating strategies were described in Appendix Fig. S10B.

### Activation-induced marker (AIM) assay

To quantify the percentages of antigen-specific Tfh cells, a total of $8 \times 10^6$ cells isolated from the spleen was plated in individual wells with RPMI supplemented with 10% FBS and 1% penicillin/streptomycin. For restimulation with protein antigen for the AIM assay, His-tagged HA protein at a concentration of 3–30 μg/mL was used. PMA/ionomycin (DAKEWE) was used as positive control, and negative control wells were included in RPMI media without protein. Cells were incubated for 24 h at 37 °C with 5% $CO_2$ before transfer to a round-bottom tube for FCM staining. Samples were incubated with the surface cocktail containing the following antibodies: FITC anti-mouse CD4 (1:200 dilution), APC/Cyanine7 anti-mouse CD3 (1:80 dilution), Brilliant Violet 421™ anti-mouse CD279 (PD-1) (1:160 dilution), APC anti-mouse CD185 (CXCR5) (1:80 dilution), PE anti-mouse CD134 (OX-40) (1:80 dilution), PE/Cyanine7 anti-mouse CD25 (1:40 dilution). The percent of AIM⁺ cells was determined by flow cytometry and the gating strategies were described in Appendix Fig. S14.

### ELISpot assay

Commercial ELISpot kits (DAKEWE) were used to measure IFN-γ and IL-4-secreting cells. Splenocytes were isolated at 1 week after booster immunization and seeded into a precoated 96-well plate. Then, cells were stimulated with purified HA protein at a concentration of 3–30 μg/mL and incubated in 5% $CO_2$ at 37 °C for 30 h. Subsequent steps were performed according to the manufacturer's instructions.

To quantify activated antibody-secreting cells (ASCs), multiscreen HA ELISpot plates (Millipore, MA, USA) were coated with purified HA protein and incubated for 16 h at 4 °C. Coated plates were washed and blocked with RPMI 1640 supplemented with 10% FBS for 2 h at 37 °C. Cell suspensions prepared from inguinal lymph nodes and bone marrow were transferred to the blocked ELISpot plates and cultured at 37 °C and 5% $CO_2$ for 24 h. The cells were lysed in ice water and incubated with biotin-IgG antibody (Bethyl Laboratories, TX, USA), streptavidin-alkaline phosphatase (Mabtech, Stockholm, Sweden) and BCIP/NBT-plus (Mabtech, Stockholm, Sweden). The plate was finally scanned, and ASC spots in each well were counted.

### Toxicity of HA@PPCDQ to DC2.4 cells

DC2.4 cells were seeded into 96-well plates with $1 \times 10^4$ cells per well. Cells were washed three times with PBS when they reached 80% confluence. And then incubated with 100 μL DMEM medium containing different concentrations of His-HA-NPs, C-His-HA-Fc-NPs, or N-His-HA-Fc-NPs for 24 h, respectively. The supernatant was removed and replaced by fresh medium containing cell counting Kit-8 (CCK-8, Abbkine). After incubation for 2 h, the viability of cells was measured by detecting absorbance at 450 nm using a SpectraMax 190 spectrophotometer (Molecular Devices, CA, USA).

### Preparation of BMDCs

Bone marrows of C57BL/6 mice were isolated and prepared into single-cell suspensions. Cells were cultured in 10% RPMI 1640 medium containing 20 ng/mL granulocyte-macrophage colony-stimulating factor (GM-CSF) and 10 ng/mL interleukin-4 (IL-4) (PeproTech, NJ, USA). Half of the medium was removed, and fresh DC medium was replenished on days 2 and 4. BMDCs were re-inoculated into 24-well plates on day 6 and stimulated with equivalent (10 μg/mL) His-HA, Fc-HA, His-HA-NPs, Fc-HA-NPs, or RPMI medium as negative control for 24 h. The cells were harvested for FCM analysis.

### Antigen uptake in vitro

DC2.4 and RAW264.7 cells were seeded into a 15 mm-cell culture dish (Biosharp) to produce a monolayer and incubated with equivalent iFlour™ 594-tagged His-HA or iFlourTM 594-tagged His-HA-NPs for 12 h. Lyso-Tracker Green (Beyotime) was diluted in DMEM and incubated with cells for 30 min at 37 °C. Nuclei were stained with Hoechst 33342 (Beyotime) for 30 min. And then imaging was performed on a Nikon A1HD25 super-resolution laser scanning confocal microscopy with a 100× NA 1.49 oil immersion objective. The mean fluorescence intensity (MFI) of HA was analyzed by ImageJ software.

### Antigen uptake in vivo

C57BL/6 mice were intramuscularly injected with equal mass of Cy5-tagged His-HA and Cy5-tagged His-HA-NPs. ILNs from both sides were harvested at 4 h post injection for living image and FCM analysis. The following antibodies were used for DC and macrophage acquisition: PE/Cyanine7 anti-CD45R/B220 (1:80 dilution), PerCP/Cyanine5.5 anti-I-A/I-E (1:300 dilution), APC/Cyanine7 anti-CD11c (1:20 dilution), PE anti-F4/80 (1:20 dilution), APC/Cyanine7 anti-CD11b (1:80 dilution), FITC anti-CD169 (1:100 dilution). The percentages of Cy5 positive DCs and macrophages were analyzed by flow cytometry.

### Antigen tracing assay

Transgenic mice CD11c-EYFP were immunized with equal mass (20 μg) of iFlour™ 594-tagged His-HA or His-HA-NPs. LNs were harvested at 4 h or day 7 post injection and fixed with 4% paraformaldehyde buffer. Prior to embedding in optimal cutting temperature compound (O.T.C., Sakura), tissues were dehydrated with 30% sucrose solution for 24 h. Next, 20 μm-thick cryosections were prepared, and images were taken on a BX63 fluorescence microscope (Olympus, Tokyo, Japan) with 4× and 10× objectives.

To monitor the residence time of PPCDQ-NPs at the injection site, ICR mice ($n = 5$ animals per group) were intramuscularly injected with equal mass (20 μg) of Cy7-tagged His-HA or His-HA-NPs. The retention time of Cy7-labeled antigen was monitored for 12 consecutive days using an IVIS Spectrum system (Caliper, Hopkinton, MA, USA). The fluorescence intensity at the injection site was semi-quantified by using Living Image 4.4 software.

To investigate the targeting ability of PPCDQ-NPs, 20 μg of Cy7-tagged His-HA or His-HA-NPs were injected into the hindlimb

**The paper explained**

**Problem**

Non-live protein subunit vaccines with a high degree of safety remain the cornerstone of modern vaccine design. However, traditional subunit vaccines using soluble proteins as antigens may still have poor immunogenicity even when formulated with adjuvants. Presenting multiple copies of an antigen in repetitive arrays has been shown to drive more robust humoral immune responses. Therefore, it is crucial to develop a versatile nanoparticle systems for multivalent display and delivery of antigens to enhance B cell responses to protein subunit vaccines.

**Results**

We constructed a vaccine adjuvant system containing Quillaja Saponaria-21 (QS-21) and cobalt porphyrin polymer micelles to achieve multivalent display of antigens on the surface of the micelles. Polymeric nanoparticles displaying repeating antigen arrays rapidly flowed to lymph nodes throughout the body and were taken up by antigen-presenting cells. This led to antigen-presenting cells (APCs) activation, further triggering a robust antigen-specific T cell immune response. When combined with QS-21, the synergy between polymeric micelles and QS-21 adjuvant induced a notable humoral immunity by facilitated the formation of germinal centers (GCs). Furthermore, the polymeric micelles prolonged the residence time of the antigen in the draining lymph nodes, and the sustained antigen availability promoted the generation of follicular helper T cells, germinal center B cells, long-lived plasma cells, and memory B cell. As a result, the polymeric micellar adjuvant system induced potent neutralizing antibodies and cytotoxic T lymphocytes that 100% protected mice against lethal influenza A virus challenge. Notably, this strategy also induced high titers of neutralizing antibodies in three other highly pathogenic viral pathogens (Ebola, Marburg, and Nipah) as well as in a rabies virus model, implying it is a promising nanovaccine system that can against multiple viruses.

**Impact**

This study presents a versatile nanovaccine platform by using the cobalt porphyrin polymer micelles and QS-21 adjuvant, which provides valuable insights for designing novel and effective antiviral vaccines against multiple pathogenic pathogens.

muscle of mice. The inguinal LNs were visualized by using an IVIS Spectrum system at 4 h post injection.

## Whole LN imaging

C57BL/6 mice were intramuscularly injected with equal mass of iFlour™ 488-tagged His-HA-NPs or His-HA. LNs were collected at 4 h post injection and immediately placed in 4% paraformaldehyde buffer (Biosharp) for 24 h. LNs were clarified via a BeyoCUBIC™ Animal Tissue Optical Clearing Kit (Beyotime) according to the manufacturer's instructions. First, the tissues were shaken at room temperature (60 rpm) with 1× Wash Buffer three times, 2 h each time. An appropriate amount of 50% BeyoCUBIC™-I Solution was added to immerse the LNs, and incubated on a shaker (60 rpm) at 37 °C for 4–24 h. Followed by 100% BeyoCUBIC™-I Solution for at least 2 days. Next, the LNs were washed three times with 1× Wash Buffer. And then placed into 50% BeyoCUBIC™-II Solution and incubated on a shaker (60 rpm) at 37 °C for 24 h. Followed by 100% BeyoCUBIC™-I Solution for 24 h and repeat this step once. Finally, the LNs were immersed in the Mounting Solution for 10 min. Imaging was performed on a Light Sheet Fluorescence Microscopy (LSFM, Zeiss Z.1) with a 5× detection objective. Snapshots and movies were generated using the Arivis Vision4D software.

## Histology

Major tissues of Mice in different immunization groups, which included heart, liver, spleen, lung, kidney, brain, intestine, and inguinal lymph nodes were collected and fixed in 4% paraformaldehyde buffer (Biosharp) for 48 h and embedded with paraffin. The brain samples were sectioned coronally and transverse sections were performed for the intestine, other tissues were sectioned longitudinally. The sections (4–5 mm) were stained with hematoxylin and eosin (H&E).

## Immunofluorescence assay

For immunofluorescence samples, OTC was used to embed. Tissue was flash-frozen, and sectioned into 20 μm slices using a CM1950 cryostat (Leica, Heerbrugg, Switzerland). Sections were blocked in PBS supplemented with 10% concentrated goat serum (Boster) for 2 h at room temperature.

The sections of lung tissue were incubated with anti-nucleoprotein antibody (NP, Influenza A, Immune technology, 1:2000 dilution) overnight at 4 °C. After washing three time, the sections were further incubated with Alexa Fluor™ 488 Goat anti-Mouse IgG (H + L) secondary antibody (1:500 dilution) for 1 h. Images were captured with a BX63 fluorescence microscope (Olympus, Tokyo, Japan).

The sections of iLN were incubated with a germinal center staining cocktail containing the following antibodies: Brilliant Violet 421™ anti-mouse/human CD45R/B220 (1:200 dilution), Alexa Fluor® 594 Goat anti-mouse IgG (1:400 dilution), and Alexa Fluor® 488 anti-mouse/human GL7 (1:200 dilution). The number of GCs was calculated according to the quantity of GL7 positive cell clusters. All images were captured with a BX63 fluorescence microscope (Olympus, Tokyo, Japan).

To observe the cross-linking between GC B cells and antigen, inguinal LNs of CD11c-EYFP mice were collected at 4 h after boost immunization with iFlour™ 594-tagged His-HA or His-HA-NPs. After cryosectioning, the slides were blocked for 2 h at room temperature in PBS buffer supplemented with 10% concentrated goat serum (Boster). Then, the cryosections were incubated overnight at 4 °C with the following antibodies: Brilliant Violet 421™ anti-mouse/human CD45R/B220 (1:200 dilution) and Alexa Fluor® 647 anti-mouse/human GL7 (1:100 dilution). Images were taken on a Nikon super-resolution spinning-disk confocal microscopy (CSU-W1, SoRa) with 10× and 20× objectives.

To observe the interactions among FDCs, GC B cells, and antigen, inguinal LNs of C57BL/6 mice were collected at 4 h after boost immunization with iFlour™ 594-tagged His-HA or His-HA-NPs. After cryosectioning, the slides were blocked and incubated with the following antibodies: BV510 anti-mouse CD35 (1:40 dilution) and Alexa Fluor® 647 anti-mouse/human GL7 (1:100 dilution) overnight at 4 °C. Images were taken on a Nikon super-resolution spinning-disk confocal microscopy (CSU-W1, SoRa) with 10× and 20× objectives.

## Statistical analysis

GraphPad Prism 8.0 software was used for all data and analyses. The log-rank (Mantel-Cox) test was used to analyze the difference in survival rate. The one-way ANOVA and two-way ANOVA were used for variance analysis of other data. Error bars for all experiments represent standard deviation (SD). The significance

of the differences in all analyses was expressed as: $*P < 0.05$; $**P < 0.01$; $***P < 0.001$, or $****P < 0.0001$.

## For more information

https://services.healthtech.dtu.dk/services/TMHMM-2.0/.

## Data availability

The datasets produced in this study are available in the following databases: Flow Cytometry: Fig. 2A. AccessionS-BSST1385. Figure 3E,F. AccessionS-BSST1384. Figure 4A. AccessionS-BSST1386. Figure 4C. AccessionS-BSST1387. Figure 4D. AccessionS-BSST1388. Figure 5D. AccessionS-BSST1389. Figure 5G. AccessionS-BSST1390. Imaging: Fig. 5J. AccessionS-BIAD1100.

The source data of this paper are collected in the following database record: biostudies:S-SCDT-10_1038-S44321-024-00076-4.

## Peer review information

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

## Acknowledgements

The authors are grateful to Dr. Zhe Hu for his support on bioimage data acquisition and Prof. Guiqing Peng for Expi 293 F cells and rVSV-ΔG-GFP plasmid. The authors sincerely thank the National Key Laboratory of Agricultural Microbiology Core Facility for assistance in flow cytometry (FCM), in vivo image system (IVIS), super-resolution laser scanning confocal microscopy (SLSCM), light sheet fluorescence microscopy (LSFM), and high-content screening system (HCSS). This work was supported by the National Key Research and Development Program of China (No. 2022YFD1800100) and the Fundamental Research Funds for the Central Universities (2662023PY005).

## Author contributions

**Caiqian Wang**: Data curation; Formal analysis; Investigation; Writing—original draft; Writing—review and editing. **Yuanyuan Geng**: Resources; Data curation; Formal analysis; Investigation. **Haoran Wang**: Data curation; Software; Formal analysis; Investigation. **Zheng Ren**: Software; Formal analysis. **Qingxiu Hou**: Data curation; Software. **An Fang**: Resources; Data curation. **Qiong Wu**: Software; Formal analysis. **Liqin Wu**: Software; Investigation. **Xiujuan Shi**:

Formal analysis; Investigation. **Ming Zhou**: Investigation; Visualization. **Zhen F Fu**: Formal analysis; Visualization. **Jonathan F Lovell**: Conceptualization; Visualization; Writing—review and editing. **Honglin Jin**: Conceptualization; Project administration; Writing—review and editing. **Ling Zhao**: Conceptualization; Funding acquisition; Project administration; Writing—review and editing.

Source data underlying figure panels in this paper may have individual authorship assigned. Where available, figure panel/source data authorship is listed in the following database record: biostudies:S-SCDT-10_1038-S44321-024-00076-4.

## Disclosure and competing interests statement

The authors declare no competing interests.

# Expanded View Figures

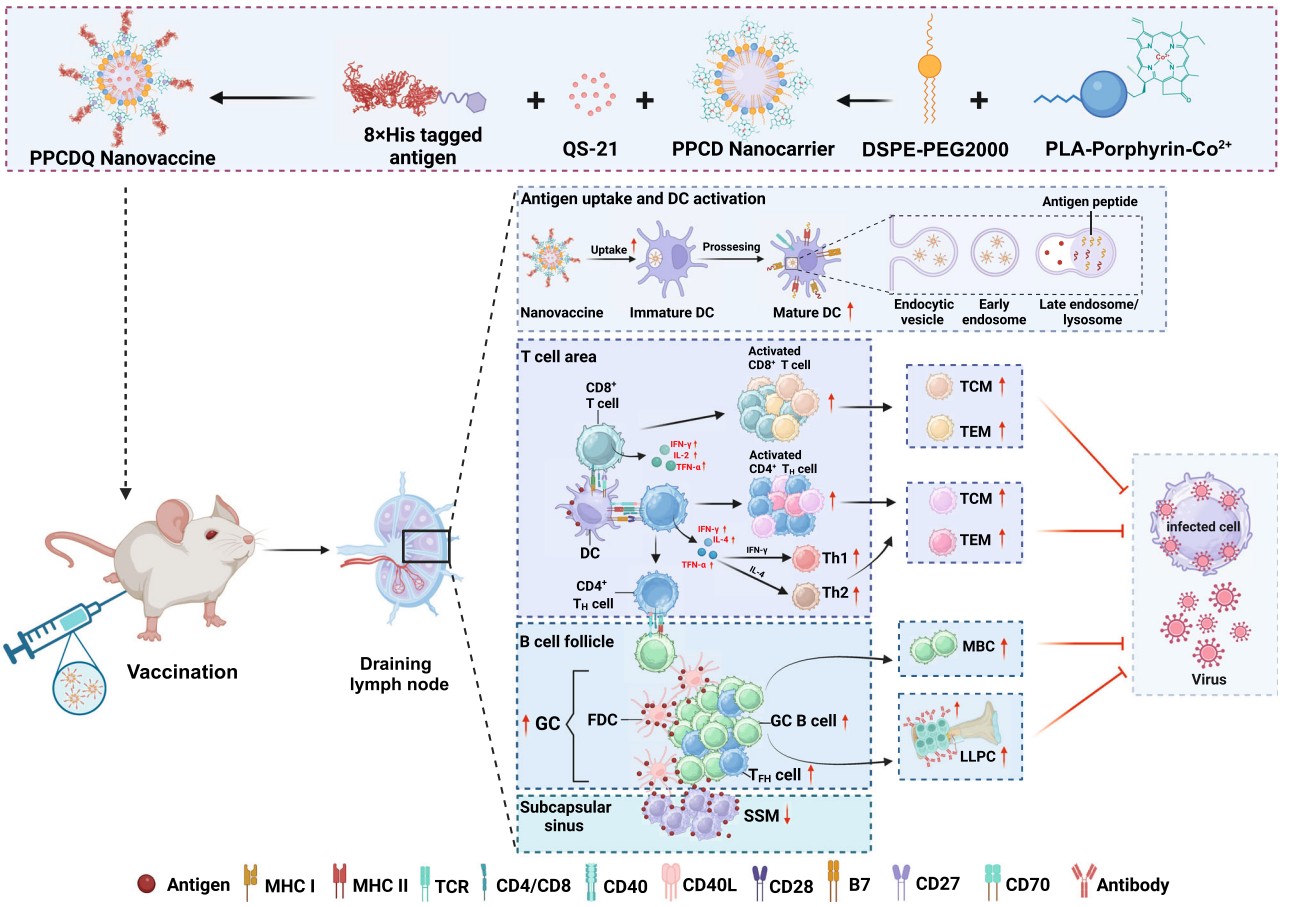

**Figure EV1. Schematic illustration of the PPCDQ nanovaccine induce a humoral and cellular immune response against virus infection.**

PPCDQ nanovaccine was synthesized by PLA-Porphyrin-Co$^{2+}$, DSPE-PEG2000, QS-21, and histidine-tagged antigen. After intramuscular injection, the PPCDQ nanovaccine drained into the lymph nodes, efficiently activated DCs and then activated T cells. Furthermore, the activated CD4$^+$ helper T cells (T$_H$ cells) activated B cells to differentiate into plasma cells (PCs) to produce antibodies. Neutralizing antibodies and cytotoxic T lymphocytes (CTLs) against virus infection together. This figure was created with BioRender (https://biorender.com/).

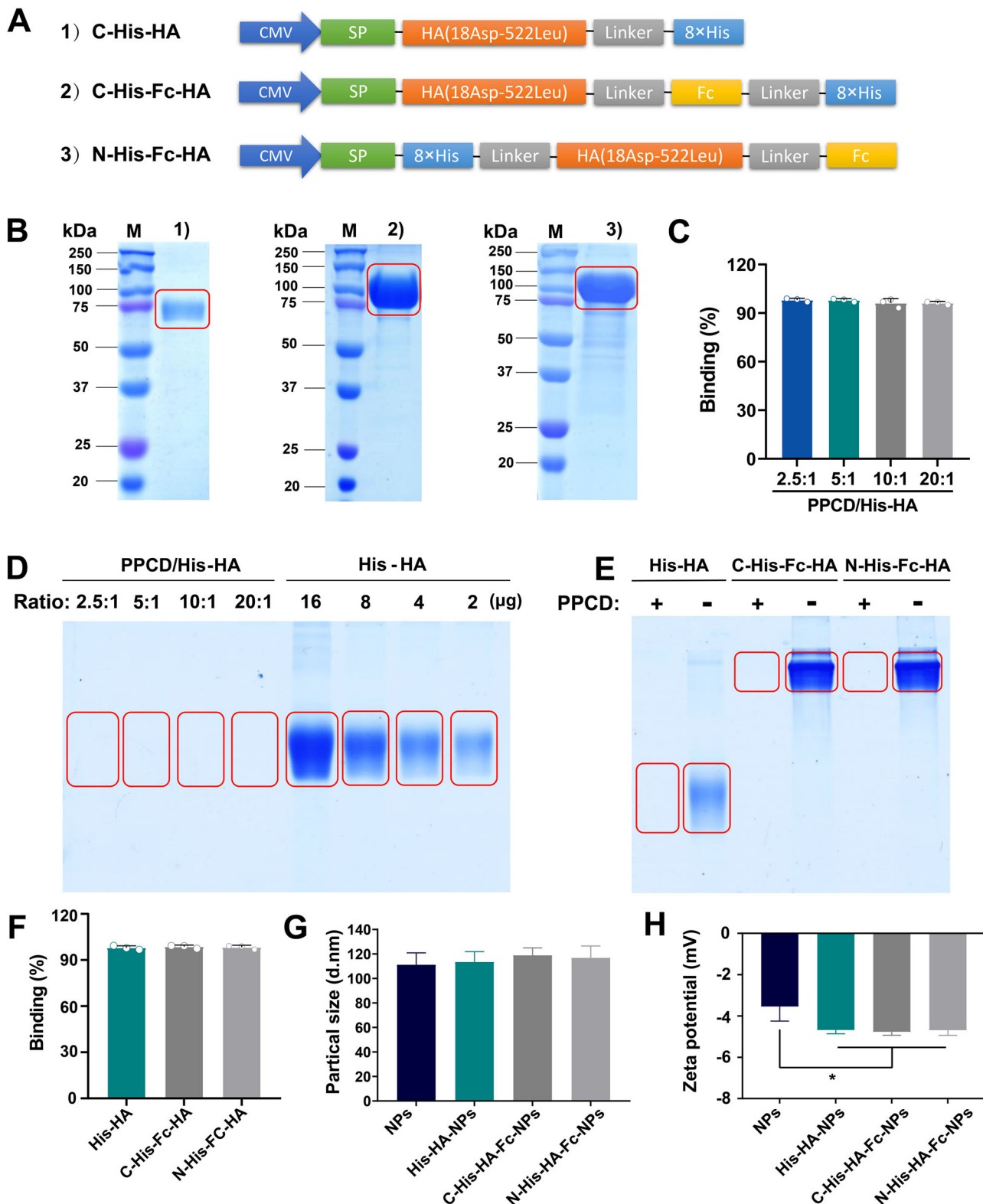

**Figure EV2.   Efficiency of PPCD anchoring His-tagged proteins.**

(A) Construction strategy of three expression plasmids. CMV was used as a promoter, an IgG1 signal peptide (SP) was fused to the N-terminal and an 8 × His tag was fused to the N-terminal or C-terminal. Fc tag was fused to the C-terminal of HA. (B) SDS-PAGE gel of C-His-HA (1), C-His-Fc-HA (2), and N-His-Fc-HA (3). (C, D) Native PAGE gel (D) and anchoring efficiency (C) of PPCD micelles with different proportions of His-HA proteins ($n = 3$ biological replicates per group). (E, F) Native PAGE gel (E) and statistical graph of binding efficiency (F) of PPCD micelles to three kinds of HA proteins ($n = 3$ biological replicates per group). (G, H) Practical size (G) and zeta potential (H) of PPCDQ combined with or without protein ($n = 3$ biological replicates per group). Data information: Data in (C, F, G, H) are mean ± SD, statistical analysis in (H) was performed by one-way ANOVA with Tukey's multiple comparisons test. *$P < 0.05$.

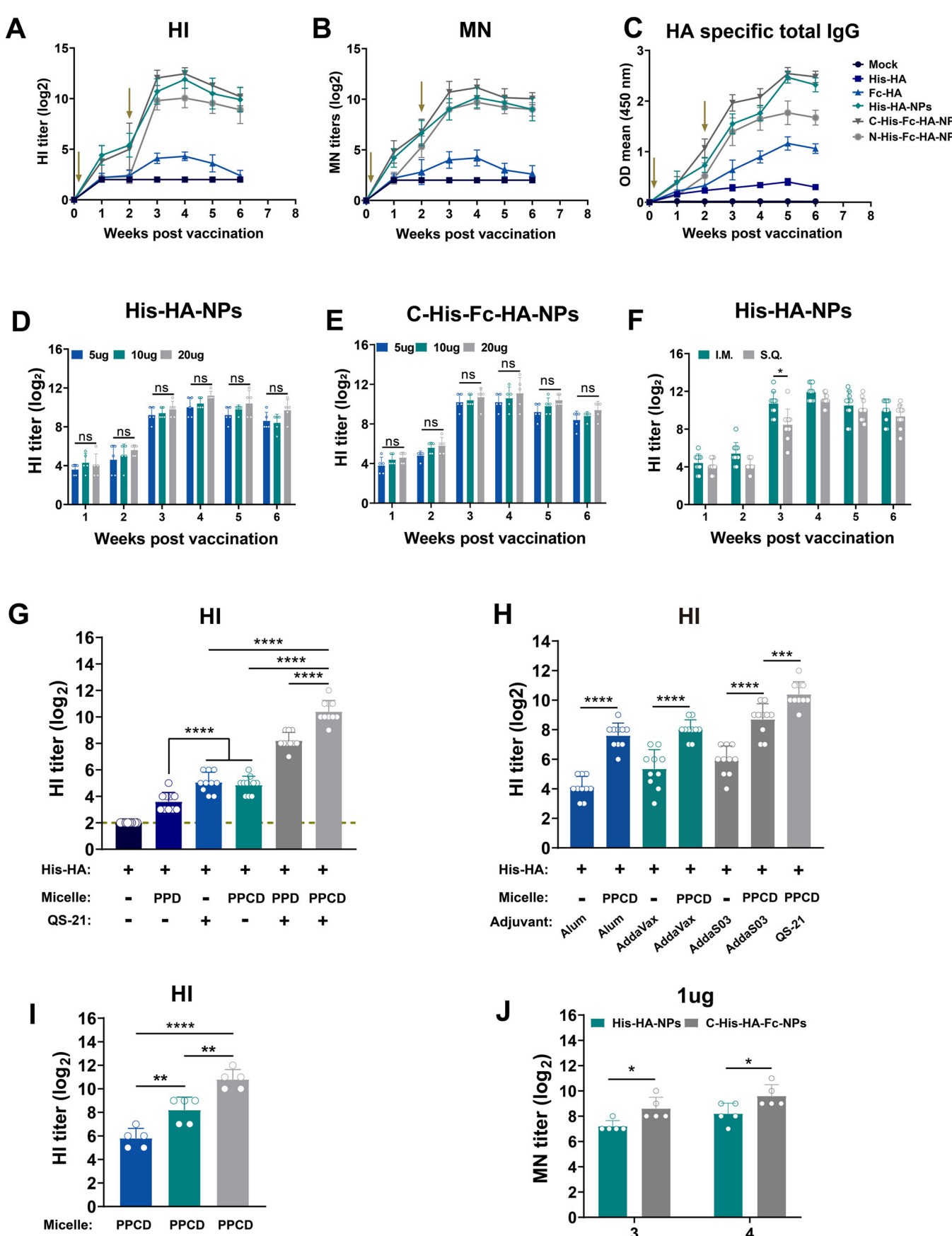

**Figure EV3.  The synergistic effect of PPCD and QS-21 is critical for antibody production.**

(A–C) HI titers (A), MN titers (B), and HA-specific total IgG (C) of immunized C57/BL6 mice at each time point ($n = 10$ animals per group). Arrows represent the time points of vaccination. (D, E) C57/BL6 mice were immunized with different dose (5, 10, and 20 μg) of His-HA-NPs (D) or C-His-Fc-HA-NPs (E). HI titers were detected at indicated time point ($n = 5$ animals per group). (F) C57/BL6 mice were intramuscularly or subcutaneously immunized with 5 μg His-HA-NPs and HI titers of each time point were tested ($n = 10$ animals per group). (G) C57/BL6 mice were immunized with different formulations and HI titers were detected at 3 weeks post vaccination ($n = 10$ animals per group). (H) C57/BL6 mice were immunized intramuscularly with the mixture of different adjuvants (Alum, AddaVax, AddaS03, and QS-21) and His-HA or His-HA@PPCD, respectively. HI titers were detected at 3 weeks post vaccination ($n = 10$ animals per group). (I) C57/BL6 mice were immunized with His-HA@PPCDQ containing different doses of QS-21(5, 10, and 20 μg). HI titers were detected at 3 weeks post vaccination ($n = 5$ animals per group). (J) C57/BL6 mice were immunized intramuscularly with 1 μg dose of His-HA-NPs or C-His-Fc-HA-NPs and MN titers were detected at 3 and 4 weeks post vaccination ($n = 5$ animals per group). Data information: Data are presented as mean ± SD, statistical analysis in (D–F, J) was determined by two-way ANOVA with Tukey's multiple comparisons test. statistical significance in (G–I) was determined by one-way ANOVA with Tukey's multiple comparisons test. $^*P < 0.05$, $^{**}P < 0.01$, $^{***}P < 0.001$, $^{****}P < 0.0001$.

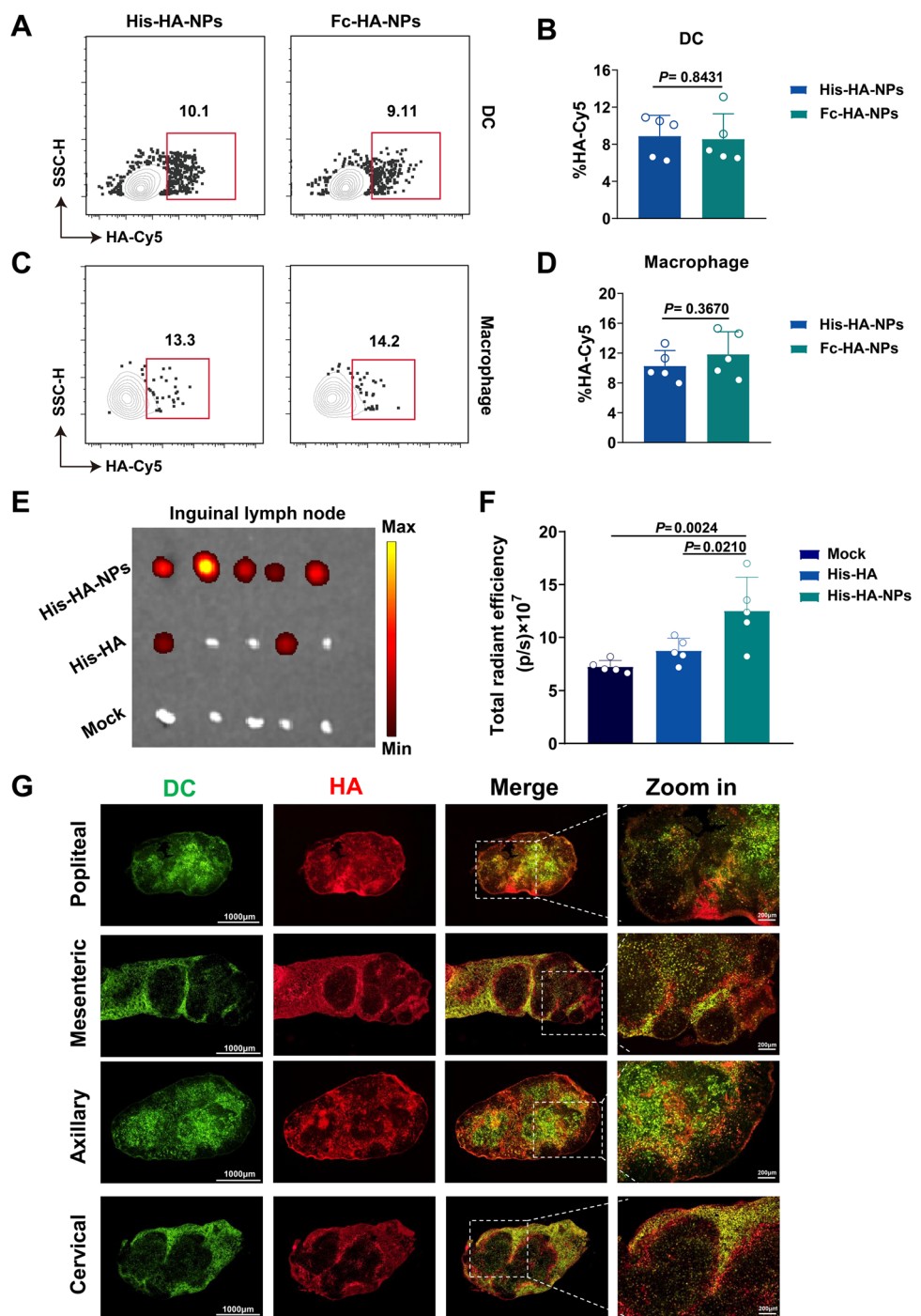

**Figure EV4. His-HA-NPs rapidly accumulate in inguinal lymph nodes and flow to remote lymph nodes.**

(A) Representative flow cytometric plots of Cy5 positive DCs (B220⁻CD11c⁺MHC⁻II⁺, top panel) in iLNs at 4 h post injection. (B) Statistical graphs of Cy5 positive DCs ($n = 5$ animals per group). (C) Representative flow cytometric plots of Cy5 positive macrophages (B220⁻CD11b⁺F4/80⁺). (D) Statistical graphs of Cy5 positive macrophages ($n = 5$ animals per group). (E) ILNs were harvested at 4 h post injection for the living image by IVIS Spectrum system. (F) Statistical graphs of the total radiant efficiency analyzed by Living Image Vision 4.4 ($n = 5$ animals per group). (G) Transgenic mice CD11c-EYFP were immunized with equal mass of iFlour™ 594-tagged His-HA or His-HA-NPs. Inguinal, popliteal, mesenteric, axillary, and cervical lymph nodes were obtained at 4 h post injection. Antigen distribution of these lymph nodes was shown by the cryosections. Scalar bar: 1000 μm (left), 200 μm (right). Data information: Data in (B, D, F) are mean ± SD, statistical analysis was determined by one-way ANOVA with Tukey's multiple comparisons test.

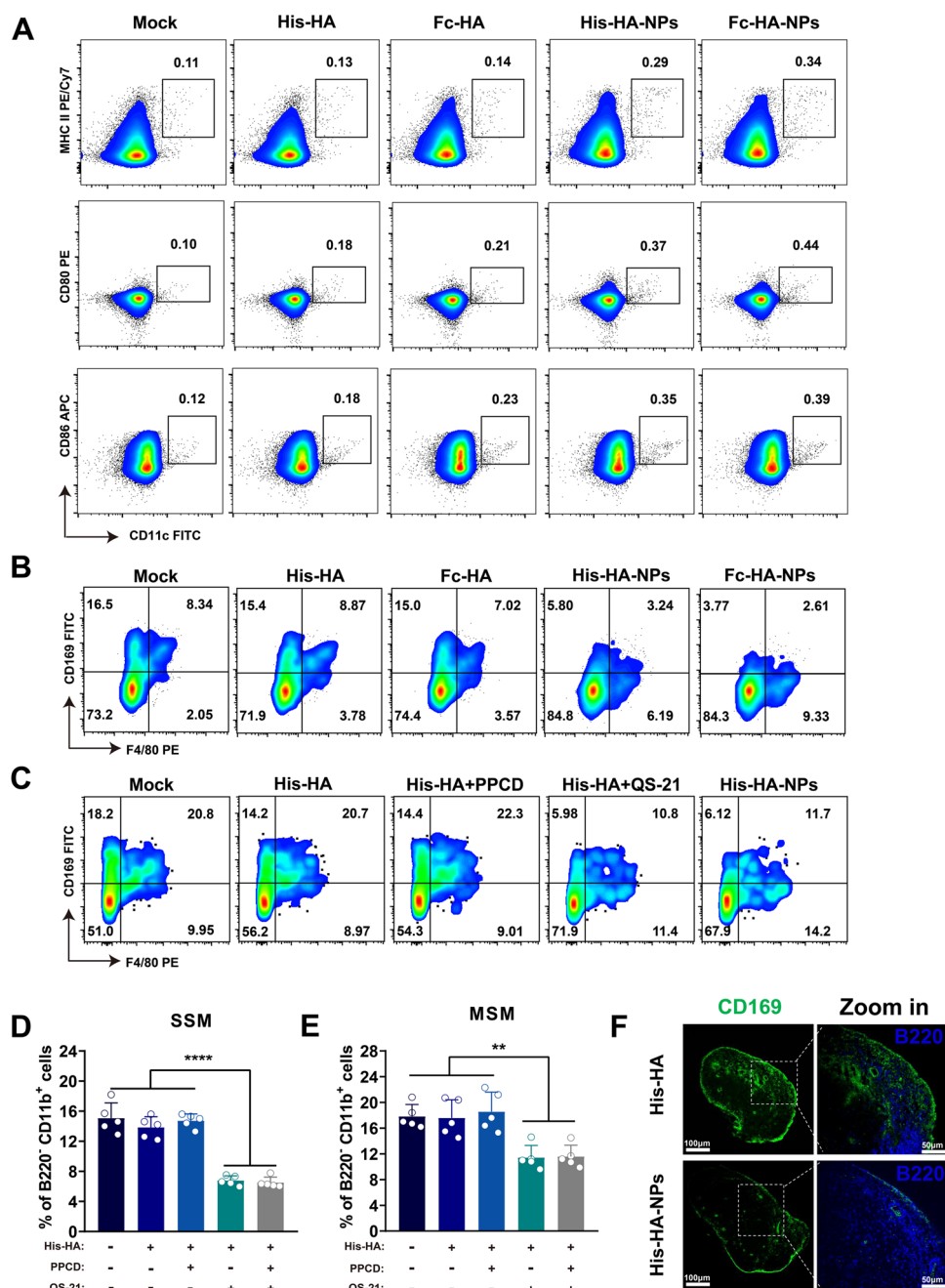

**Figure EV5. HA@PPCDQ promotes DC activation and depletes CD169⁺ subcapsular sinus macrophages.**

(A) Representative flow cytometric plots of CD11c⁺MHC-II⁺DCs, CD11c⁺CD80⁺ DCs, and CD11c⁺CD86⁺DCs in iLNs at 2 days post boost immunization. (B) Representative flow cytometric plots of subcapsular sinus macrophage (SSM, B220⁻CD11b⁺F4/80⁻CD169⁺) and medullary sinus macrophage (MSM, B220⁻CD11b⁺F4/80⁺CD169⁺) in iLN. (C–E) C57/BL6 mice were immunized with different formulations, and iLNs were collected for FCM. (C) Representative flow cytometric plots of SSM and MSM. Statistical graphs of SSM (D) and MSM (E) (n = 5 animals per group). (F) Cryosections of inguinal lymph nodes were incubated with FITC anti-mouse CD169 (Siglec-1) and Brilliant Violet 421™ anti-mouse/human CD45R/B220. Images were taken on a Nikon super-resolution spinning-disk confocal microscopy. Scalar bar: 100 μm (left) and 50 μm (right). Data information: Data in (D), and (E) are mean ± SD, statistical analysis was determined by one-way ANOVA with Tukey's multiple comparisons test. **P < 0.01, ****P < 0.0001.

