## [Peer Review File · EMBO Molecular Medicine]

A Broadly Applicable Protein-Polymer Adjuvant System for Antiviral Vaccines

Ling Zhao, Caiqian Wang, Yuanyuan Geng, Haoran Wang, Zeheng Ren, Qingxiu Hou, An Fang, Qiong Wu, Liqin Wu, Xiujuan Shi, Ming Zhou, Zhen F. Fu, Jonathan Lovell, and Honglin Jin

Corresponding authors: Ling Zhao (lingzhao@mail.hzau.edu.cn), Honglin Jin (jin@hust.edu.cn), Jonathan Lovell (jflorell@buffalo.edu)

Review Timeline:

Submission Date:	5th Dec 23
Editorial Decision:	11th Jan 24
Revision Received:	4th Apr 24
Editorial Decision:	18th Apr 24
Revision Received:	23rd Apr 24
Accepted:	25th Apr 24

Editor: *Zeljko Durdevic*

Transaction Report:

11th Jan 2024

Dear Prof. Zhao,

Thank you for the submission of your manuscript to EMBO Molecular Medicine. We have now received feedback from the three reviewers who agreed to evaluate your manuscript. All three referees recognize potential interest of the study but also raise important and partially overlapping criticism that should be addressed in a major revision. If you would like to discuss further the points raised by the referees, I am available to do so via email or video. Let me know if you are interested in this option.

We would welcome the submission of a revised version within three months for further consideration. Please let us know if you require longer to complete the revision.

I look forward to receiving your revised manuscript.

Yours sincerely,

Zeljko Durdevic

We require:

- 1) A .docx formatted version of the manuscript text (including legends for main figures, EV figures and tables). Please make sure that the changes are highlighted to be clearly visible.
- 2) Individual production quality figure files as .eps, .tif, .jpg (one file per figure). For guidance, download the 'Figure Guide PDF': (<https://www.embopress.org/page/journal/17574684/authorguide#figureformat>).
- 3) A .docx formatted letter INCLUDING the reviewers' reports and your detailed point-by-point responses to their comments. As part of the EMBO Press transparent editorial process, the point-by-point response is part of the Review Process File (RPF), which will be published alongside your paper.
- 4) A complete author checklist, which you can download from our author guidelines (<https://www.embopress.org/page/journal/17574684/authorguide#submissionofrevisions>). Please insert information in the checklist that is also reflected in the manuscript. The completed author checklist will also be part of the RPF.
- 5) Please note that all corresponding authors are required to supply an ORCID ID for their name upon submission of a revised manuscript.
- 6) It is mandatory to include a 'Data Availability' section after the Materials and Methods. Before submitting your revision, primary datasets produced in this study need to be deposited in an appropriate public database, and the accession numbers and

database listed under 'Data Availability'. Please remember to provide a reviewer password if the datasets are not yet public (see <https://www.embopress.org/page/journal/17574684/authorguide#dataavailability>).

13) Author contributions: You will be asked to provide CRediT (Contributor Role Taxonomy) terms in the submission system. These replace a narrative author contribution section in the manuscript.

14) A Conflict of Interest statement should be provided in the main text.

15) Every published paper now includes a 'Synopsis' to further enhance discoverability. Synopses are displayed on the journal

webpage and are freely accessible to all readers. They include a short stand first (maximum of 300 characters, including space) as well as 2-5 one-sentences bullet points that summarizes the paper. Please write the bullet points to summarize the key NEW findings. They should be designed to be complementary to the abstract - i.e. not repeat the same text. We encourage inclusion of key acronyms and quantitative information (maximum of 30 words / bullet point). Please use the passive voice. Please attach these in a separate file or send them by email, we will incorporate them accordingly.

Please also suggest a striking image or visual abstract to illustrate your article as a PNG file 550 px wide x 300-800 px high.

**** Reviewer's comments ****

Referee #1 (Remarks for Author):

In this present study, the authors developed a cobalt-based polymeric micelles for direct conjugation of His-tag antigens for vaccination. QS-21 is incorporated as an adjuvant for immune stimulation, and three different hemagglutinin recombinant proteins were assessed in the study. The article is notable for having multiple layers of immunological characterizations, including lymph node and immune cell targeting, humoral immunity, cellular immunity, and immune cell recruitment. In addition, the authors used five viral proteins from different pathogenic viruses to verify the effectiveness of this vaccine system, fully demonstrating its broad applicability. Overall, this is an impressive manuscript with complete logic, and the results and conclusions mostly support the data. The manuscript would be improved if the authors addressed the following issues:

Major points

1. It is recommended to place the synthesis schematic and some characterization data in EV2 and EV3 to Figure 1, which helps to demonstrate the idea of vaccine design.
2. How to purify the synthesized NPs? What is the storage temperature and stabilization time? Please provide above detailed information.
3. Figure EV2: The cobalt-free micelle controls are lacking to verify that PPCDQ anchors His-tagged proteins via Co²⁺.
4. Figures 3D and Lines 484-486: The authors speculate that subcapsular sinus macrophages (SSM) depletion is caused by the QS-21 adjuvant in the PPCDQ, but there is a lack of PPCD micelle control to exclude the impact of nanocarriers on SSM.
5. Figure 3E and G: The authors found that the number of spleen neutrophils increased significantly after HA@PPCDQ immunization. Please explain the possible reasons for this phenomenon and its physiological significance in the discussion section.
6. Figure 4H: It is recommended to further analyze the proportion of IL-2+TNF α +IFN γ +, IL-2+TNF α +IFN γ -, and IL-2+TNF α -IFN γ - T cells, etc. And their correlation with antibody titers.
7. Figure EV15: It is hard to see clearly the co-localization between red and rose red. Please change the pseudo colors with stronger contrast.
8. The description of the transmission electron microscopy (TEM) method is too simple, and please improve it.
9. Please supplement the relevant primer sequences for plasmid construction in Materials and Methods or in the supplemental materials.

Mini points

1. Lines 130 :What is the abbreviation PLA? The full name should be written when it appears for the first time.
2. Lines 133-135: Lengthy sentences, it is recommended to delete "with vaccination".
3. Lines 138-139: Revise the sentence structure to "verified with the antigens of.....virus".
4. Lines 269-272: Since the lymph nodes drained by different immune methods are different, it is recommended to change "drain" to "flow".
5. Lines 472-473: The authors missed the immune effect of PPCDQ against three other highly pathogenic pathogens.
6. Line 843: Toxicity of "PPCDQ " to DC2.4 cells? The subtitle does not match the content, please revise.

Referee #2 (Comments on Novelty/Model System for Author):

The study presents comparison of three different histidine-tagged hemagglutinin variants, demonstrates superiority of the particulate adjuvant over several conventional adjuvants, and provides in-depth examination of the particulate vaccine's distribution, immune activation, as well as the resulting cellular and humoral immunity. In addition, the study demonstrates the utility of the particulate adjuvants on multiple viral antigens. Particularly notable in the study is the employment of lattice sheet microscopy for lymph node examination and the exploration of varying levels of subcapsular sinus macrophages, which offers new insights to the field of nanoparticulate vaccines.

Referee #2 (Remarks for Author):

In the article "A Broadly Applicable Protein-Polymer Adjuvant System for Antiviral Vaccines," the authors present a Quillaja Saponaria-21 (QS-21) polymeric micelle functionalized with cobalt porphyrin as a particulate adjuvant for direct association with histidine-tagged antigens. The study presents comparison of three different histidine-tagged hemagglutinin variants, demonstrates superiority of the particulate adjuvant over several conventional adjuvants, and provides in-depth examination of the particulate vaccine's distribution, immune activation, as well as the resulting cellular and humoral immunity. In addition, the study demonstrates the utility of the particulate adjuvants on multiple viral antigens. Particularly notable in the study is the employment of lattice sheet microscopy for lymph node examination and the exploration of varying levels of subcapsular sinus macrophages, which offers new insights to the field of nanoparticulate vaccines. Overall the article is well written and is of broad interest to the readers of EMBO molecular medicine. It is recommended for publication upon resolution of the following comments.

1. The primary novelty of the work is to demonstrate that the cobalt-porphyrin-based HIS-tag antigen association approaches, which has been reported previously with liposomal systems, can be employed on a polymeric micelle. In this regard, it should be noted that porphyrin is a highly hydrophobic molecule that has a high likelihood to be buried inside the inaccessible core of the micelle. As polymeric micelles tend to have high surface energy that can result in antigen adsorption, the authors should perform a control antigen study with non his-tagged HA, which can help confirm cobalt/his-tag mediated association.
2. Given the high antibody titers raised by the particulate vaccine against HA, the weight loss under viral challenge in Figure 6 and Figure EV16 seems a tad disappointing. HA-based vaccine strategies under a prime-boost regimen typically leads to near-sterilizing protectivity in literature. Do the authors have any thought on the weight loss? Could it be altered epitope/antigenicity associated with the particulate vaccine?

Referee #3 (Comments on Novelty/Model System for Author):

It would be helpful to know how many animals were used in different groups to determine the significance and biological relevance of the results and the justification for the number of animals used, etc.

Referee #3 (Remarks for Author):

The authors of this manuscript assessed a new vaccine formulation system comprising of the Quillaja Saponaria-21(QS-21) and cobalt porphyrin polymeric lipid micelles to present his-tagged protein antigens on the surface of the micelle. They showed that these engineered nanoscale micelles could promote protein antigen's uptake and dendritic cell activation to induce a robust cytotoxic T lymphocyte response and to mediate germinal center formation. When recombinant protein antigens from influenza A and rabies virus were used in this system, they could elicit robust antiviral responses that protected mice from lethal challenge by the respective viruses. Additionally, the authors showed that these micelles could be combined with other viral antigens of Ebola virus, Marburg virus, and Nipah virus to induce high titers of neutralizing antibodies against these highly pathogenic viral pathogens.

Overall, this is a very comprehensive study with a significant level of novelty and significance in addressing multiple aspects of this new vaccine delivery platform. The experimental designs and executions were reasonably well done. The authors tested this novel nanocarrier by conducting multiple experiments to understand how the immune system responds to its immunization and validated its efficacy in protecting from lethal challenges in animal models. However, the writing is a bit dense with many technical and scientific jargons and assumptions that might not be ideal (or suitable) for non-specialists who are the targeted audience of this journal (per the journal's review guidelines). The manuscript can be polished further with attentions paid to the correct ordering of the figures as they are being described in the text. In other words, there are several errors in describing the different figures, which make it difficult to follow. It would be helpful for the authors to provide some rationales and justifications for some of the experiments that were being performed. Some of these deficiencies are outlined below for the authors to consider when revising the manuscript. It would also be helpful to know how many animals were used in different groups to determine the potential significance of data.

1. The authors compared different commercial adjuvants in their study and demonstrated a higher level of effectiveness of PPCDQ. However, the different commercial adjuvants were used without PPCD, unlike QS-21, which was combined with the nanocarrier (PPCDQ). It would be great if other commercial adjuvants can also be mixed with PPCD and compared in order to increase the significance of the finding.
2. In Fig 2. The authors immunized mice with Cy7-tagged His-HA-NPs or an equivalent amount of Cy7-tagged HA. In parallel in vivo results were further verified by antigen uptake experiments in vitro using DC2.4 and RAW264.7 cells. The authors need to provide a rationale for not using Fc-HA-NPs in this particular in vivo/in vitro study despite the fact that this modification was

shown to elicit higher antibody titers.

3. In Fig EV 5G, the authors demonstrated the effectiveness of multiple modified delivery platforms, including PPCD, PPD, and PPCDQ. Interestingly, PPD with QS-21 elicits higher HI titers than free antigens with QS-21. The authors need to provide a rationale or an explanation for this as it appears PPD should not conjugate with His tagged antigen as the cobalt is missing. Is the nanoparticle (PPD) and antigen supposed to function separately in this specific modification? Or does the PPD alone also induce some degree of an immune response?

4. The authors mentioned that the results in Fig. 4A and B showed that MHCII, CD80, and CD86 molecules were significantly upregulated after HA@PPCDQ treated BMDCs for 12 h. Do they mean to say Fig. 3A and B instead?

5. The authors also mentioned Fig 5F and G stating IFN-gamma producing cells increased about 7-fold in the HA@PPCDQ group...Do they mean Fig 4F and G instead?

6. The authors referred to Fig. 6J and L to show the number of GCs in the HA@PPCDQ group was 4-8 times higher than that of the His-HA or Fc-HA group. Do they mean Fig 5J and L instead?

7. The authors didn't mention the number of mice used in each group for the immunization and challenge study. It would be helpful to know how many animals were used in different groups to determine the significance and biological relevance of the results.

Point Response Letter

Dear Editor,

Thank you for your decision letter concerning our manuscript titled " *A Broadly Applicable Protein-Polymer Adjuvant System for Antiviral Vaccines* " (EMM-2023-19093), and your time regarding for our revision. I also appreciate all the critical comments from you and the reviewers. We have carefully considered the comments and revised the manuscript accordingly. With these improvements, we hope that the current version will meet the Journal's standards for publication. The following is a point-by-point response to all the comments and a list of changes we have made to the manuscript.

Sincerely,

Ling Zhao, PhD

Point-by-point responses to the comments of the reviewer and a list of changes are:

Response to Referee #1

Referee #1 (Remarks for Author):

In this present study, the authors developed a cobalt-based polymeric micelles for direct conjugation of His-tag antigens for vaccination. QS-21 is incorporated as an adjuvant for immune stimulation, and three different hemagglutinin recombinant proteins were assessed in the study. The article is notable for having multiple layers of immunological characterizations, including lymph node and immune cell targeting, humoral immunity, cellular immunity, and immune cell recruitment. In addition, the authors used five viral proteins from different pathogenic viruses to verify the effectiveness of this vaccine system, fully demonstrating its broad applicability. Overall, this is an impressive manuscript with complete logic, and the results and conclusions mostly support the data. The manuscript would be improved if the

authors addressed the following issues:

Major points

1. It is recommended to place the synthesis schematic and some characterization data in EV2 and EV3 to Figure 1, which helps to demonstrate the idea of vaccine design.

Response: Thanks for the suggestion. We have added a synthesis schematic to Fig 1 and incorporated synthesis-related characterization data into the Fig EV2 to better illustrate the ideal of vaccine design.

2. How to purify the synthesized NPs? What is the storage temperature and stabilization time? Please provide above detailed information.

Response: Thanks for the comments. The synthesized NPs were purified by the size-exclusion chromatography (SEC) and stored at 4 °C. We have added this detail in the Materials and Methods section (line 680-682). In addition, we tested the stabilization time of PPCDQ micelle pre-bound with His-Fc-HA protein at 4 °C. As the results are shown below, Fc-HA-NPs can be stored at 4 °C for at least three weeks and maintain immunogenicity comparable to freshly prepared micelle. We have added these data in the Appendix Figure S18 (Page 19).

Response Figure 1. Stability of PPCDQ micelle pre-bound with C-His-Fc-HA.

3. Figure EV2: The cobalt-free micelle controls are lacking to verify that PPCDQ anchors His-tagged proteins via Co^{2+} .

Response: Thanks for the comments. Following your suggestion, we tested the efficiency of cobalt-free micelle (PPDQ) for anchoring His-HA proteins and used cobalt-containing micelles (PPCDQ) as a control. The result of native PAGE showed that PPDQ micelle was unable to anchor His-HA proteins, indicating that PPCDQ anchors proteins by cobalt interaction with His tags. We've added these information in the Results section (line 174-178, Appendix Figure S3)

Response Figure 2. His-HA anchoring efficacy in different formulations.

4. Figures 3D and Lines 484-486: The authors speculate that subcapsular sinus macrophages (SSM) depletion is caused by the QS-21 adjuvant in the PPCDQ, but there is a lack of PPCD micelle control to exclude the impact of nanocarriers on SSM.

Response: Thank you for the constructive suggestion. In order to further verify whether the nanocarrier PPCD can affect subcapsular macrophages (SSM), we immunized mice with His-HA, His-HA+QS-21, His-HA+PPCD, and His-HA+PPCD+QS-21(His-HA-NPs), respectively. The data of flow cytometry showed that immunization with His-HA or His-HA+PPCD did not lead to SSM depletion, while immunization with His-HA+QS-21 and His-HA-NPs led to the same degree of SSM depletion. This result well verifies our speculation in the discussion section: SSM depletion is caused by the QS-21 adjuvant in the PPCDQ. We have

refined this result in the manuscript (Fig EV5C-E, line 331-335 and line 524-526).

Response Figure 3. QS-21 adjuvant in PPCDQ causes subcapsular sinus macrophage (SSM) depletion.

5. Figure 3E and G: The authors found that the number of spleen neutrophils increased significantly after HA@PPCDQ immunization. Please explain the possible reasons for this phenomenon and its physiological significance in the discussion section.

Response: Thanks for the comments. We have added content about neutrophils to the Discussion section (line 538-548) as follows:

“Additionally, QS-21-induced NLRP3 inflammasome activation may be responsible for the recruitment of neutrophils to the spleen after HA@PPCDQ immunization. Because circulating neutrophils are rapidly recruited to sites of inflammation in response to infectious and inflammatory stimuli (Kastenmüller, Torabi-Parizi et al., 2012, Lok & Clatworthy, 2021). Meanwhile, CD169⁺ macrophages, Caspase-1 and MyD88 deficiencies have been shown to affect the recruitment of neutrophils to lymph nodes after QS-21 immunization (Detienne, Welsby et al., 2016). The recruited neutrophils may regulate the subsequent adaptive

immunity induced by HA@PPCDQ in many aspects, such as by interacting with DCs for antigen presentation(Bennouna & Denkers, 2005, Lok & Clatworthy, 2021), activating B cells(Balázs, Martin et al., 2002, Kolaczkowska & Kubes, 2013), promoting T cell immune response(Kesteman, Vansanten et al., 2008, Tillack, Breiden et al., 2012), etc.”

Reference

Kastenmüller W, Torabi-Parizi P, Subramanian N, Lämmermann T, Germain RN. A spatially-organized multicellular innate immune response in lymph nodes limits systemic pathogen spread. *Cell* 2012;150(6):1235-1248.

Lok LSC, Clatworthy MR. Neutrophils in secondary lymphoid organs. *Immunology* 2021;164(4):677-688.

Detienne S, Welsby I, Collignon C, Wouters S, Coccia M et al. Central Role of CD169(+) Lymph Node Resident Macrophages in the Adjuvanticity of the QS-21 Component of AS01. *Scientific reports* 2016;6:39475.

Bennouna S, Denkers EY. Microbial antigen triggers rapid mobilization of TNF-alpha to the surface of mouse neutrophils transforming them into inducers of high-level dendritic cell TNF-alpha production. *J Immunol* 2005;174(8):4845-4851.

Balázs M, Martin F, Zhou T, Kearney J. Blood dendritic cells interact with splenic marginal zone B cells to initiate T-independent immune responses. *Immunity* 2002;17(3):341-352.

Kolaczkowska E, Kubes P. Neutrophil recruitment and function in health and inflammation. *Nature reviews Immunology* 2013;13(3):159-175.

Kesteman N, Vansanten G, Pajak B, Goyert SM, Moser M. Injection of lipopolysaccharide induces the migration of splenic neutrophils to the T cell area of the white pulp: role of CD14 and CXC chemokines. *Journal of leukocyte biology* 2008;83(3):640-647.

Tillack K, Breiden P, Martin R, Sospedra M. T lymphocyte priming by neutrophil extracellular traps links innate and adaptive immune responses. *J Immunol*

6. Figure 4H: It is recommended to further analyze the proportion of IL-2⁺TNFα⁺IFNγ⁺, IL-2⁺TNFα⁺IFNγ⁻,and IL-2⁺TNFα⁻IFNγ⁻ T cells, etc. And their correlation with antibody titers.

Response: Thanks for the comments. We analyzed the proportions of various populations which secreting cytokines in CD4⁺ T cells, and further explored the relationship between populations and HA specific IgG antibodies. As shown in the figure below, there is a strong correlation between TNF-α⁺IL-2⁺IFN-γ⁻ cells and antibody titers in Fc-HA, His-HA-NPs, and Fc-HA-NPs group. We have added these data in the Result section (line 382-384).

Response Figure 4. PPCDQ elicits robust CD4⁺ T cell responses.

7. Figure EV15: It is hard to see clearly the co-localization between red and rose red. Please change the pseudo colors with stronger contrast.

Response: Thanks for the suggestion. In order to better distinguish the two colors, we replaced the pseudo color of HA with gray and re-merged it. As shown in the figure below (Appendix Figure S15, Page 21).

Response Figure 5. HA@PPCDQ facilitates cross-linking between antigen and GC B cells.

8. The description of the transmission electron microscopy (TEM) method is too simple, and please improve it.

Response: Thanks for the suggestion. We have added more detailed operating procedures of transmission electron microscopy (TEM) in the Materials and Methods section (line 691-694) as follow:

“Purified HA@PPCDQ samples were diluted to 100 $\mu\text{g}/\text{ml}$ in PBS, and 5 μL of each was placed on 200 copper grids for 45 s, respectively. The grids were washed twice with 5 μl of distilled water. Excess liquid was removed and samples were air-dried. Grids were imaged on an HT7700 (Hitachi, Japan) microscope at 100.0 kV.”

9. Please supplement the relevant primer sequences for plasmid construction in Materials and Methods or in the supplemental materials.

Response: Thanks for the comments. We have compiled the relevant primers for plasmid construction in the table shown below and attached them in the Appendix Table S1(Page 21).

TABLE 1 Primers used for plasmid construction.

Primer	Sequences (5'-3')
His-HA-F	GGCGTGCAGAGTGACACCATCTGCATCGGGTACC
His-HA-R	CGCCGCCTCCCAGCTTCACTCCGTCCACCTTCTC
Fc-HA-F	TTCATCATTTTGGCAAAGAATTCGCCACCATGGACTGGACCT
Fc-HA-R	GTACATATGCAAGGCTTACAACCCTCGAGACTTCCTCCTCCTC
C-His-Fc-HA-F	ATCATTTTGGCAAAGAATTCGCCACCATGGACTGGA
C-His-Fc-HA-R	ACTTCTCCTCCTCCGCTGCCGCCGCCTCCTTTACCAGGAGAGTGGGAG
N-His-Fc-HA-F	ATCACCATCACCATCACCATCACGGAAGCGGAAGCGACACCATCTGCATC
N-His-Fc-HA-R	CCGCTGCCGCCGCCTCCCAGCTTCACTCCGTCCAC
NIPA-F	CAGCGGAGGAGGAGGAAGTCAAACTACACAAGATCAAC
NIPA-R	AAAAGATCTGCTAGCTCGAGTTATGTACATTGCTCTGGTATCTT
EBOV-RBD-F	GTGGCCGCCGCCACAGGCGTGCAGAGTATCCCACTTGGAGTCATCCAC
EBOV-RBD-R	GATGACTTCCTCCTCCTCCGCTGCCGCCGCCTCCGAAAGACAACCTTCA
MARV-GP1-F	TTCATCATTTTGGCAAAGAATTCGCCACCATGAAGACAACCTGTCTGTT
MARV-GP1-R	TCCTCCTCCGCTGCCGCCGCCTCCTCTTTTCTTTCTGAAGTAGAC

Mini points

1. Lines 130 :What is the abbreviation PLA? The full name should be written when it appears for the first time.

Response: Thanks for the suggestion. We have added the full name of poly(lactic acid) (PLA) in the introduction section (line 94-95) and checked the abbreviations throughout the text.

2. Lines 133-135: Lengthy sentences, it is recommended to delete "with vaccination".

Response: Thank you for the correction. We have modified this sentence to: "Furthermore, we evaluated the effectiveness of this nano-platform in mice using the hemagglutinin (HA) protein of influenza A virus (IAV) and the rabies virus (RABV) glycoprotein (RABV-G) as models." (line 136-138)

3. Lines 138-139: Revise the sentence structure to "verified with the antigens of.....virus".

Response: Thanks for the suggestion. The sentence was corrected to: "The universality of the PPCDQ nanopatform was further verified with the antigens of

Ebola virus (EBOV), Marburg virus (MARV) and Nipah virus (NiV).” (line 141-143)

4. Lines 269-272: Since the lymph nodes drained by different immune methods are different, it is recommended to change "drain" to "flow".

Response: Thanks for the comment. As suggested, we have revised "drain" to "flow". (line 291).

5. Lines 472-473: The authors missed the immune effect of PPCDQ against three other highly pathogenic pathogens.

Response: Thanks for the guidance. We have improved this sentence to: “ Like other saponin-containing liposome (such as AS01, Matrix M or other ISCOMs), the PPCDQ encapsulating QS-21 significantly increased the immunogenicity of IAV, EBOV, MARV, NiV, and RABV subunit vaccines.” (line 510-511)

6. Line 843: Toxicity of "PPCDQ " to DC2.4 cells? The subtitle does not match the content, please revise.

Response: Thank you for the correction again. We have changed the subtitle “Toxicity of His-HA-NPs to DC2.4 cells” into “Toxicity of HA@PPCDQ to DC2.4 cells” (line 905).

Response to Referee #2

Referee #2 (Comments on Novelty/Model System for Author):

The study presents comparison of three different histidine-tagged hemagglutinin variants, demonstrates superiority of the particulate adjuvant over several conventional adjuvants, and provides in-depth examination of the particulate vaccine's distribution, immune activation, as well as the resulting cellular and humoral immunity. In addition, the study demonstrates the utility of the particulate adjuvants on multiple viral antigens. Particularly notable in the study is the employment of

lattice sheet microscopy for lymph node examination and the exploration of varying levels of subcapsular sinus macrophages, which offers new insights to the field of nanoparticulate vaccines.

Response: We sincerely appreciate the Referee #2 very positive comments and extremely constructive suggestions.

Referee #2 (Remarks for Author):

In the article "A Broadly Applicable Protein-Polymer Adjuvant System for Antiviral Vaccines," the authors present a Quillaja Saponaria-21 (QS-21) polymeric micelle functionalized with cobalt porphyrin as a particulate adjuvant for direct association with histidine-tagged antigens. The study presents comparison of three different histidine-tagged hemagglutinin variants, demonstrates superiority of the particulate adjuvant over several conventional adjuvants, and provides in-depth examination of the particulate vaccine's distribution, immune activation, as well as the resulting cellular and humoral immunity. In addition, the study demonstrates the utility of the particulate adjuvants on multiple viral antigens. Particularly notable in the study is the employment of lattice sheet microscopy for lymph node examination and the exploration of varying levels of subcapsular sinus macrophages, which offers new insights to the field of nanoparticulate vaccines. Overall, the article is well written and is of broad interest to the readers of EMBO molecular medicine. It is recommended for publication upon resolution of the following comments.

Response: We thank Referee #2 for a nice summary of our work and all the positive comments.

1. The primary novelty of the work is to demonstrate that the cobalt-porphyrin-based HIS-tag antigen association approaches, which has been reported previously with liposomal systems, can be employed on a polymeric micelle. In this regard, it should be noted that porphyrin is a highly hydrophobic molecule that has a high likelihood to be buried inside the inaccessible core of the micelle. As polymeric micelles tend to have high surface energy that can result in antigen adsorption, the authors should

perform a control antigen study with non his-tagged HA, which can help confirm cobalt/his-tag mediated association.

Response: Thanks for the constructive comment. Following the suggestion, we prepared cobalt-free micelle PPDQ and Fc-HA protein without His-tag (purified by Protein A+G resin). Furthermore, we tested the efficiency of anchoring proteins with two different formulations (PPDQ+His-HA and PPCDQ+Fc-HA) using native PAGE assay. As shown in the figure below, cobalt-deficient PPDQ micelles cannot anchor His-HA protein, and PPCDQ cannot capture Fc-HA protein lacking his-tag. In addition, the antibody production results after immunization with different formulas also indirectly proved that the conjugation between PPCDQ micelles and antigens is mediated by cobalt and His- tag.

Although porphyrins are highly hydrophobic, the results of transmission electron microscopy (TEM) and native PAGE assay demonstrated that the position of the Porphyrin-Co²⁺ did not affect the access of the His-tag as well as the binding of the antigen. Indeed, our previous study showed that capture of His-tagged proteins by cobalt porphyrin phospholipids (Co-PoP) occurs within a protected hydrophobic bilayer, resulting in essentially irreversible attachment in serum. Moreover, this combination is related to the length of His-tag (Shao, Geng et al., 2015, Figure.4b). Therefore, we chosen the 8-histidine tagged proteins to ensure its binding to PPCDQ micelle in the present study.

Antigen adsorption due to the high surface energy of polymer micelles is possibly nonspecific and unstable(Sapsford, Algar et al., 2013). Our results showed that the weak binding between the Fc-HA protein and PPCDQ micelle was abolished after 4 hours of electrophoresis at 120V, indicating that the stable conjugation between antigens and micelles may require other modifications. We have added these information in the Results section (line 174-178 and line 234-236, Appendix Figure S3).

Response Figure 6. The conjugation between PPCDQ micelles and antigens is mediated by cobalt and His-tag.

Reference

Shao S, Geng J, Ah Yi H, Gogia S, Neelamegham S, Jacobs A, Lovell JF (2015) Functionalization of cobalt porphyrin–phospholipid bilayers with his-tagged ligands and antigens. *Nature Chemistry* 7: 438-446

Sapsford KE, Algar WR, Berti L, Gemmill KB, Casey BJ et al. Functionalizing nanoparticles with biological molecules: developing chemistries that facilitate nanotechnology. *Chemical reviews* 2013;113(3):1904-2074.

2. Given the high antibody titers raised by the particulate vaccine against HA, the weight loss under viral challenge in Figure 6 and Figure EV16 seems a tad disappointing. HA-based vaccine strategies under a prime-boost regimen typically leads to near-sterilizing protectivity in literature. Do the authors have any thought on

the weight loss? Could it be altered epitope/antigenicity associated with the particulate vaccine?

Response: Thanks for the critical comment. We think there are two main reasons for the weight loss after viral challenge.

First, the stress response may be responsible for the weight loss in mice after challenge. Because we used a high-dose, mouse-adapted IAV strain PR8 (A/Puerto Rico/8/1934 [H1N1]) for challenge test in the present study.

Secondly, it is possibly related to the administration methods of immunization and challenge. In our study, all mice were immunized intramuscularly and then challenged intranasally. Intramuscular immunization usually fails to induce mucosal immune responses and produce antigen-specific IgG and IgA in nasal and lung, resulting in the inability to block viruses in the upper respiratory tract or lungs immediately after challenge. A small amount of virus replication in the lungs may be responsible for the weight loss of mice in the first few days after challenge. In fact, the body weight loss post challenge has been observed in other studies of influenza vaccines administered intramuscularly or subcutaneously (Hendin, Lavoie et al., 2022, Sharma, Zhang et al., 2024, Sia, He et al., 2021).

As you speculated, it is indeed possible to increase the protectivity of influenza vaccines by altering the antigen epitopes presented on the nanoparticle vaccine. Previous studies have shown that nanovaccines displaying the conserved stem region of HA protein can generate broadly protective antibody against heterosubtypic influenza virus (Impagliazzo, Milder et al., 2015, Yassine, Boyington et al., 2015). In addition, an intranasal multivalent nanoparticles tandemly presenting conserved epitopes of three influenza virus proteins have been reported to induce high levels of antibody and cellular immune responses, and provide complete protection against lethal infection with multiple influenza viruses (Pan, Wang et al., 2023). However, in this study, we focused on testing the usability and immune mechanism of the PPCDQ nanoplatform and did not perform epitope optimization. In future studies, we will optimize epitopes to mobilize more effective immune responses to improve the protection of influenza vaccines.

Reference

Hendin HE, Lavoie PO, Gravett JM, Pillet S, Saxena P et al. Elimination of receptor binding by influenza hemagglutinin improves vaccine-induced immunity. *NPJ vaccines* 2022;7(1):42.

Sharma P, Zhang X, Ly K, Zhang Y, Hu Y et al. The lipid globotriaosylceramide promotes germinal center B cell responses and antiviral immunity. *Science (New York, NY)* 2024;383(6684):eadg0564.

Sia ZR, He X, Zhang A, Ang JC, Shao S et al. A liposome-displayed hemagglutinin vaccine platform protects mice and ferrets from heterologous influenza virus challenge. *Proceedings of the National Academy of Sciences of the United States of America* 2021;118(22).

Impagliazzo A, Milder F, Kuipers H, Wagner MV, Zhu X et al. A stable trimeric influenza hemagglutinin stem as a broadly protective immunogen. *Science (New York, NY)* 2015;349(6254):1301-1306.

Yassine HM, Boyington JC, McTamney PM, Wei CJ, Kanekiyo M et al. Hemagglutinin-stem nanoparticles generate heterosubtypic influenza protection. *Nature medicine* 2015;21(9):1065-1070.

Pan J, Wang Q, Qi M, Chen J, Wu X et al. An Intranasal Multivalent Epitope-Based Nanoparticle Vaccine Confers Broad Protection against Divergent Influenza Viruses. *ACS nano* 2023;17(14):13474-13487.

Response to Referee #3

Referee #3 (Comments on Novelty/Model System for Author):

It would be helpful to know how many animals were used in different groups to determine the significance and biological relevance of the results and the justification for the number of animals used, etc.

Response: Thank you for the reminder and correction. We have added the number of animals used in each group to the legend section of all figures.

Referee #3 (Remarks for Author):

The authors of this manuscript assessed a new vaccine formulation system comprising of the Quillaja Saponaria-21(QS-21) and cobalt porphyrin polymeric lipid micelles to present his-tagged protein antigens on the surface of the micelle. They showed that these engineered nanoscale micelles could promote protein antigen's uptake and dendritic cell activation to induce a robust cytotoxic T lymphocyte response and to mediate germinal center formation. When recombinant protein antigens from influenza A and rabies virus were used in this system, they could elicit robust antiviral responses that protected mice from lethal challenge by the respective viruses. Additionally, the authors showed that these micelles could be combined with other viral antigens of Ebola virus, Marburg virus, and Nipah virus to induce high titers of neutralizing antibodies against these highly pathogenic viral pathogens.

Overall, this is a very comprehensive study with a significant level of novelty and significance in addressing multiple aspects of this new vaccine delivery platform. The experimental designs and executions were reasonably well done. The authors tested this novel nanocarrier by conducting multiple experiments to understand how the immune system responds to its immunization and validated its efficacy in protecting from lethal challenges in animal models. However, the writing is a bit dense with many technical and scientific jargons and assumptions that might not be ideal (or suitable) for non-specialists who are the targeted audience of this journal (per the journal's review guidelines). The manuscript can be polished further with attentions paid to the correct ordering of the figures as they are being described in the text. In other words, there are several errors in describing the different figures, which make it difficult to follow. It would be helpful for the authors to provide some rationales and justifications for some of the experiments that were being performed. Some of these deficiencies are outlined below for the authors to consider when revising the manuscript. It would also be helpful to know how many animals were used in different groups to determine the potential significance of data.

Response: We appreciate Referee #3 for this great summary of our work and the

constructive comments. Following the suggestions, we have improved the manuscript writing and added the basic justifications and corresponding reasons for the experiments that were performed (e.g., line 264-267, line 299-301, line 311-313, line 351-354, line 403-405, et al.). Additionally, we corrected the descriptions of the figure order in the results section and supplemented the number of animals used in each group in the figure legends.

1. The authors compared different commercial adjuvants in their study and demonstrated a higher level of effectiveness of PPCDQ. However, the different commercial adjuvants were used without PPCD, unlike QS-21, which was combined with the nanocarrier (PPCDQ). It would be great if other commercial adjuvants can also be mixed with PPCD and compared in order to increase the significance of the finding.

Response: Thanks for the comment. As suggested, we further tested the immune effects of three commercial adjuvants mixed with PPCD, and used His-HA-NPs as a control. As shown in the figure below, compared to the mixture with HA monomers, all three commercial adjuvants mixed with HA@PPCD induced higher HI titers. However, the antibody titers were still significantly lower than those of His-HA-NPs, approximately three times lower. Due to the commercial adjuvant itself is in the form of nanoparticles (according to the manufacturer's instructions), it may not be able to integrate with PPCD to function in the same peripheral lymph nodes or cells. In addition, the nanoformulation of saponin adjuvant has been shown to have advantages over other commercial adjuvants in antigen uptake and promoting B cell proliferation and activation (Silva, Kato et al., 2021). This may be the reason why the immunological effects of His-HA-NPs are greatly enhanced when QS-21 is incorporated into PPCD micelles. We have added these data in the Results section (Fig EV3G, line 243-248).

Response Figure 7. Immunity effects of different formulation.

Reference

Silva M, Kato Y, Melo MB, Phung I, Freeman BL et al. A particulate saponin/TLR agonist vaccine adjuvant alters lymph flow and modulates adaptive immunity. *Sci Immunol* 2021;6(66):eabf1152.

2. In Fig 2. The authors immunized mice with Cy7-tagged His-HA-NPs or an equivalent amount of Cy7-tagged HA. In parallel in-vivo results were further verified by antigen uptake experiments in vitro using DC2.4 and RAW264.7 cells. The authors need to provide a rationale for not using Fc-HA-NPs in this particular in vivo/in vitro study despite the fact that this modification was shown to elicit higher antibody titers.

Response: Thanks for the comment. In Fig. 2, the reason that we used His-HA-NPs for antigen tracing is because there is no difference between the two micelles in lymph node targeting, DC and macrophage uptake *in vivo*. We re-validated this result using two Cy5-labeled antigens (Cy5-His-HA and Cy5-His-Fc-HA). As shown in the figure below, the results of vivo imaging and flow cytometry showed there is no significant difference between the two micelles. Therefore, we only chosen His-HA and His-HA-NPs for subsequent experiments, focusing on comparing the differences

between monovalent and multivalent antigens in lymph node accumulation, APC uptake, and presentation. We have described these data in the Results section (line 260-267) and presented them in the Fig EV4A-D and Appendix Figure S5.

Response Figure 8. Antigen uptake in vivo is comparable between the His-HA-NPs and Fc-HA-NPs.

3. In Fig EV 5G, the authors demonstrated the effectiveness of multiple modified delivery platforms, including PPCD, PPD, and PPCDQ. Interestingly, PPD with QS-21 elicits higher HI titers than free antigens with QS-21. The authors need to provide a rationale or an explanation for this as it appears PPD should not conjugate with His tagged antigen as the cobalt is missing. Is the nanoparticle (PPD) and antigen supposed to function separately in this specific modification? Or does the PPD alone also induce some degree of an immune response?

Response: Thanks for the nice comment. To test this speculation, we evaluated the

immune effect of PPD+His-HA (PPD micelle and His-HA antigen function separately) and compared it with other formulations. Surprisingly, immunization with a mixture of His-HA and PPD micelles induced detectable anti-HA antibodies (HI titer > 4). As you speculated, PPD micelles may also induce some degree of immune response. Previous research has shown nanoparticles can not only be used as carriers and platforms to deliver drugs, but also as adjuvants to directly mix with antigens for vaccination (Alameh, Tombácz et al., 2021, Nguyen & Tolia, 2021, Zhao, Seth et al., 2014). Polymeric nanoparticles have also been shown in some studies to directly serve as adjuvants to stimulate immune responses (Luo, Wang et al., 2017, Noh, Hong et al., 2013). Combined with our results, we speculate that PPD micelles may have inherent immunogenicity to stimulate some degree of immune response. (line 226-229)

Furthermore, QS-21 adjuvant exerted a stronger immune response when formulated with PPD micelles, which is possibly another reason why His + PPD + QS-21 elicited higher HI titers than the free antigen of QS-21. Previous studies have shown that QS-21 (ISCOM, AS01 and Matrix) integrated in liposomes induces strong adaptive immune responses to co-administered antigens and that it does not matter whether the antigen is conjugated to ISCOM (Cox, Pedersen et al., 2011, Fries, Smith et al., 2013, Heath, Galiza et al., 2021, Pedersen, Madhun et al., 2012, Silva et al., 2021). In this study, the free QS-21 adjuvant did not trigger a strong immune response, but when formulated with micelles, the immune effect was greatly enhanced. It is speculated that the combination of QS-21 and PPD micelles promotes its stability and helps it flow to lymph nodes faster to perform its effects. To sum up, the immune stimulating effect of PPD micelles itself plus the QS-21 delivered by PPD micelles are the main reasons for the enhanced immune responses.

Response Figure 9. Immune effects of different formulations.

Reference

Alameh MG, Tombácz I, Bettini E, Lederer K, Sittplangkoon C et al. Lipid nanoparticles enhance the efficacy of mRNA and protein subunit vaccines by inducing robust T follicular helper cell and humoral responses. *Immunity* 2021;54(12):2877-2892.e2877.

Nguyen B, Tolia NH. Protein-based antigen presentation platforms for nanoparticle vaccines. *NPJ vaccines* 2021;6(1):70.

Zhao L, Seth A, Wibowo N, Zhao CX, Mitter N et al. Nanoparticle vaccines. *Vaccine* 2014;32(3):327-337.

Luo M, Wang H, Wang Z, Cai H, Lu Z et al. A STING-activating nanovaccine for cancer immunotherapy. *Nature nanotechnology* 2017;12(7):648-654.

Noh YW, Hong JH, Shim SM, Park HS, Bae HH et al. Polymer nanomicelles for efficient mucus delivery and antigen-specific high mucosal immunity. *Angewandte Chemie (International ed in English)* 2013;52(30):7684-7689.

Cox RJ, Pedersen G, Madhun AS, Svindland S, Sævik M et al. Evaluation of a

virosomal H5N1 vaccine formulated with Matrix M™ adjuvant in a phase I clinical trial. *Vaccine* 2011;29(45):8049-8059.

Fries LF, Smith GE, Glenn GM. A recombinant viruslike particle influenza A (H7N9) vaccine. *The New England journal of medicine* 2013;369(26):2564-2566.

Heath PT, Galiza EP, Baxter DN, Boffito M, Browne D et al. Safety and Efficacy of NVX-CoV2373 Covid-19 Vaccine. *The New England journal of medicine* 2021;385(13):1172-1183.

Pedersen GK, Madhun AS, Breakwell L, Hoschler K, Sjursen H et al. T-helper 1 cells elicited by H5N1 vaccination predict seroprotection. *The Journal of infectious diseases* 2012;206(2):158-166.

Silva M, Kato Y, Melo MB, Phung I, Freeman BL et al. A particulate saponin/TLR agonist vaccine adjuvant alters lymph flow and modulates adaptive immunity. *Sci Immunol* 2021;6(66):eabf1152.

4. The authors mentioned that the results in Fig. 4A and B showed that MHCII, CD80, and CD86 molecules were significantly upregulated after HA@PPCDQ treated BMDCs for 12 h. Do they mean to say Fig. 3A and B instead?

Response: Thank you for the correction, we have accurate the order of this figure in the text.

5. The authors also mentioned Fig 5F and G stating IFN-gamma producing cells increased about 7-fold in the HA@PPCDQ group...Do they mean Fig 4F and G instead?

Response: Thanks for the suggestions and we have corrected it.

6. The authored referred to Fig. 6J and L to show the number of GCs in the HA@PPCDQ group was 4-8 times higher than that of the His-HA or Fc-HA group. Do they mean Fig 5J and L instead?

Response: Thank you for the correction again, we have done it.

7. The authors didn't mention the number of mice used in each group for the immunization and challenge study. It would be helpful to know how many animals were used in different groups to determine the significance and biological relevance of the results.

Response: Thank you for your reminder and correction. We have added the number of animals used in each group to the legend section of all figures.

18th Apr 2024

Dear Prof. Zhao,

Thank you for the submission of your revised manuscript to EMBO Molecular Medicine. I am pleased to inform you that we will be able to accept your manuscript pending the following final amendments:

- 1) We note that you currently have together with you, a total of 3 co-corresponding authors. Is that correct? Do you confirm equal contribution of these 3 people, able to take full responsibility for the paper and its content? While there is no limit per se to the number of co-corresponding authors, 3 is rare, and may not reflect as intended to the community.
- 2) Figures: We note that in Appendix Figures 7 and 10 the same dataset is used please indicate this in their figure legends.
- 3) In the main manuscript file, please do the following:
 - Please address all comments suggested by our data editors listed below:
 - o Figure legends:
 1. Please note that the legends for figures 2b-d is not provided in the sequential manner (legend for figures 2c-d are provided before legend of figure 2b). This needs to be rectified.
 2. Please note that the legends for figures 5k-l is not provided in the sequential manner (legend for figure 5l is provided before legend of figure 5k). This needs to be rectified.
 3. Please note that in figures 1f-h; 3b-d, g-h; 7b-c, e-f, h-i, k-l; EV 3h; there is a mismatch between the annotated p values in the figure legend and the annotated p values in the figure file that should be corrected.
 - In Methods section, provide the antibody dilutions that were used for each antibody.
 - Please include structured Methods section that includes a Reagents and Tools Table followed by a Methods and Protocols section. Please check "Author Guidelines" for more information and to download table templates.
<https://www.embopress.org/page/journal/17574684/authorguide#structuredmethods>
 - Move "Ethics statements" to the Methods section.
 - Author contributions: Please remove it from the manuscript and specify author contributions in our submission system. CRediT has replaced the traditional author contributions section because it offers a systematic machine-readable author contributions format that allows for more effective research assessment. You are encouraged to use the free text boxes beneath each contributing author's name to add specific details on the author's contribution. More information is available in our guide to authors:
<https://www.embopress.org/page/journal/17574684/authorguide#authorshipguidelines>
 - In data availability section please remove "The published article includes all datasets for generating main figures during this study. Source data uploaded to BioStudies:" and information about the figures. Use the following format to report the accession number of your data:

The datasets produced in this study are available in the following databases:
[data type]: [full name of the resource] [accession number/identifier] ([doi or URL or identifiers.org/DATABASE:ACCESSION])
- Please check "Author Guidelines" for more information.
<https://www.embopress.org/page/journal/17574684/authorguide#availabilityofpublishedmaterial>
- Please move the reference list at the end of the manuscript file after "For more information". Also, correct the reference citation in the reference list. Where there are more than 10 authors on a paper, 10 will be listed, followed by "et al.". Please check "Author Guidelines" for more information.
<https://www.embopress.org/page/journal/17574684/authorguide#referencesformat>
- 4) Synopsis: Please check your synopsis text and image before submission with your revised manuscript. Please be aware that in the proof stage minor corrections only are allowed (e.g., typos).
- 5) Source Data: Source data for Figure 3F are missing. Please clarify if these data are already contained in source data of another figure and indicate it in the source data checklist, if this is not the case, please provide source data for the Figure 3F.
- 6) As part of the EMBO Publications transparent editorial process initiative (see our Editorial at <http://embomolmed.embopress.org/content/2/9/329>), EMBO Molecular Medicine will publish online a Review Process File (RPF) to accompany accepted manuscripts. This file will be published in conjunction with your paper and will include the anonymous referee reports, your point-by-point response and all pertinent correspondence relating to the manuscript. Let us know whether you agree with the publication of the RPF and as here, if you want to remove or not any figures from it prior to publication. Please note that the Authors checklist will be published at the end of the RPF.
- 7) Please provide a point-by-point letter INCLUDING my comments as well as the reviewer's reports and your detailed responses (as Word file).

I look forward to reading a new revised version of your manuscript as soon as possible.

Yours sincerely,

Zeljko Durdevic

*** Instructions to submit your revised manuscript ***

- 1) a .docx formatted version of the manuscript text (including Figure legends and tables)
- 2) Separate figure files*
- 3) supplemental information as Expanded View and/or Appendix. Please carefully check the authors guidelines for formatting Expanded view and Appendix figures and tables at <https://www.embopress.org/page/journal/17574684/authorguide#expandedview>
- 4) a letter INCLUDING the reviewer's reports and your detailed responses to their comments (as Word file).
- 5) The paper explained: EMBO Molecular Medicine articles are accompanied by a summary of the articles to emphasize the major findings in the paper and their medical implications for the non-specialist reader. Please provide a draft summary of your article highlighting
 - the medical issue you are addressing,
 - the results obtained and
 - their clinical impact.This may be edited to ensure that readers understand the significance and context of the research. Please refer to any of our published articles for an example.
- 6) For more information: There is space at the end of each article to list relevant web links for further consultation by our readers. Could you identify some relevant ones and provide such information as well? Some examples are patient associations, relevant databases, OMIM/proteins/genes links, author's websites, etc...
- 7) Author contributions: the contribution of every author must be detailed in a separate section.
- 8) EMBO Molecular Medicine now requires a complete author checklist (<https://www.embopress.org/page/journal/17574684/authorguide>) to be submitted with all revised manuscripts. Please use the checklist as guideline for the sort of information we need WITHIN the manuscript. The checklist should only be filled with page numbers where the information can be found. This is particularly important for animal reporting, antibody dilutions (missing) and exact values and n that should be indicated instead of a range.
- 9) Every published paper now includes a 'Synopsis' to further enhance discoverability. Synopses are displayed on the journal webpage and are freely accessible to all readers. They include a short stand first (maximum of 300 characters, including space) as well as 2-5 one sentence bullet points that summarise the paper. Please write the bullet points to summarise the key NEW findings. They should be designed to be complementary to the abstract - i.e. not repeat the same text. We encourage inclusion of key acronyms and quantitative information (maximum of 30 words / bullet point). Please use the passive voice. Please attach

these in a separate file or send them by email, we will incorporate them accordingly.

You are also welcome to suggest a striking image or visual abstract to illustrate your article. If you do please provide a jpeg file 550 px-wide x 300-800px high.

10) A Conflict of Interest statement should be provided in the main text

11) Please note that we now mandate that all corresponding authors list an ORCID digital identifier. This takes <90 seconds to complete. We encourage all authors to supply an ORCID identifier, which will be linked to their name for unambiguous name identification.

Currently, our records indicate that the ORCID for your account is 0000-0003-0569-8105.

Link Not Available

Photos 400-800 DPI

*Additional important information regarding figures and illustrations can be found at

<https://bit.ly/EMBOPressFigurePreparationGuideline>. See also figure legend preparation guidelines:

<https://www.embopress.org/page/journal/17574684/authorguide#figureformat>

***** Reviewer's comments *****

Referee #1 (Remarks for Author):

All concerns are addressed.

Referee #3 (Remarks for Author):

Is suitable for publication.

The authors addressed the minor editorial issues.

25th Apr 2024

Dear Prof. Zhao,

We are pleased to inform you that your manuscript is accepted for publication and is now being sent to our publisher to be included in the next available issue of EMBO Molecular Medicine.
